# RAD51 restricts DNA over-replication from re-activated origins

Sergio Muñoz [ID][1], Elena Blanco-Romero[1,5], Daniel González-Acosta[1,2,5], Sara Rodriguez-Acebes [ID][1], Diego Megías[3,4], Massimo Lopes [ID][2] & Juan Méndez [ID][1✉]

## Abstract

**Eukaryotic cells rely on several mechanisms to ensure that the genome is duplicated precisely once in each cell division cycle, preventing DNA over-replication and genomic instability. Most of these mechanisms limit the activity of origin licensing proteins to prevent the reactivation of origins that have already been used. Here, we have investigated whether additional controls restrict the extension of re-replicated DNA in the event of origin re-activation. In a genetic screening in cells forced to re-activate origins, we found that re-replication is limited by RAD51 and enhanced by FBH1, a RAD51 antagonist. In the presence of chromatin-bound RAD51, forks stemming from re-fired origins are slowed down, leading to frequent events of fork reversal. Eventual re-initiation of DNA synthesis mediated by PRIMPOL creates ssDNA gaps that facilitate the partial elimination of re-duplicated DNA by MRE11 exonuclease. In the absence of RAD51, these controls are abrogated and re-replication forks progress much longer than in normal conditions. Our study uncovers a safeguard mechanism to protect genome stability in the event of origin reactivation.**

**Keywords** DNA Replication; Fork Progression; MRE11; RAD51; Re-replication
**Subject Categories** Cell Cycle; DNA Replication, Recombination & Repair

## Introduction

In mammalian cells, tens of thousands of DNA replication origins are activated in the S phase to start genomic duplication. Each origin must be 'fired' only once to prevent DNA over-replication and chromosomal damage (reviewed by Alexander and Orr-Weaver, 2016). Origin re-firing is minimized by the temporal separation of licensing and activation: ORC, CDC6, and CDT1 proteins attract the MCM helicase to form pre-replicative complexes (pre-RCs) in G1, whereas CDK and DDK kinases

promote full replisome assembly and DNA synthesis in S phase (reviewed by Limas and Cook, 2019). Following origin activation, re-licensing is restricted by overlapping mechanisms (Thakur et al, 2022). At least three pre-RC components (ORC1, CDC6, and CDT1) are targeted by ubiquitin ligases (Méndez et al, 2002; Tatsumi et al, 2003; Arias and Walter, 2006; Nishitani et al, 2006; Walter et al, 2016). In addition, CDT1 is inhibited by CDK phosphorylation (Zhou et al, 2020) and binding to geminin (GMN; Wohlschlegel et al, 2000). The importance of the latter is underscored by the fact that GMN depletion is sufficient to promote origin re-licensing and re-firing (Zhu et al, 2004; Klotz-Noack et al, 2012).

In budding yeast, origin re-firing may drive gene amplification (Green et al, 2010; Finn and Li, 2013; Hanlon and Li, 2015). In mammalian cells, over-replication generates ssDNA gaps and double-strand breaks (DSBs; Davidson et al, 2006; Neelsen et al, 2013), activating the G2/M checkpoint and promoting apoptosis (Zhu et al, 2004; Muñoz et al, 2017). In mice, overexpression of CDC6 and CDT1 causes over-replication in precursor stem cells, leading to lethal tissue dysplasia (Muñoz et al, 2017). DNA over-replication has also been linked to aggressive behavior in cancer cells (Galanos et al, 2016; Petropoulos et al, 2023).

Despite its deleterious consequences, origin re-licensing and re-firing might not be infrequent under physiological conditions (Dorn et al, 2009; Reusswig and Pfander, 2019; Reusswig et al, 2022). Multiple MCM complexes are engaged at early origins (Das et al, 2015), where most induced DNA re-replication takes place (Menzel et al, 2020; Fu et al, 2021). Some origins may undergo repetitive rounds of activation and give rise to up to short (<200 bp) fragments of re-replicated DNA (Gómez and Antequera, 2008). Besides, ~2% of the forks detected in stretched DNA fibers display patterns consistent with reactivated origins (Dorn et al, 2009; Muñoz et al, 2017). This situation is exacerbated in cancer cells (Dorn et al, 2009) in which pre-RC proteins are frequently overexpressed (reviewed by Petrakis et al, 2016). In all cases, the extent of DNA over-replication must be contained to be compatible with survival.

Forks established on newly-replicated DNA (from now on, "re-replication forks") display lower processivity and are more prone to collapse into DSBs than regular forks (Green et al, 2006; Tanny

[1]DNA Replication Group, Molecular Oncology Programme, Spanish National Cancer Research Centre (CNIO), Melchor Fernández Almagro 3, 28029 Madrid, Spain. [2]Institute of Molecular Cancer Research, University of Zurich, Winterthurerstrasse 190, 8057 Zurich, Switzerland. [3]Confocal Microscopy Unit, Biotechnology Programme, Spanish National Cancer Research Centre (CNIO), Melchor Fernández Almagro 3, 28029 Madrid, Spain. [4]Advanced Optical Microscopy Unit, Central Core Facilities, Instituto de Salud Carlos III, Madrid, Spain. [5]These authors contributed equally: Elena Blanco-Romero, Daniel González-Acosta. ✉E-mail: jmendez@cnio.es

et al, 2006; Finn and Li, 2013; Muñoz et al, 2017; Fu et al, 2021). In yeast, standard forks may synthesize up to 100–200 kb of DNA (Newlon et al, 1993; van Brabant et al, 2001) while re-replication forks rarely progress more than 30–35 kb (Nguyen et al, 2001). Mammalian re-replication forks are also slower and the mean size of over-replicated DNA stretches is 58 kb (Fu et al, 2021). These antecedents suggest that specific mechanisms restrict the extension of re-replicated DNA after origin re-firing.

Here we report a role for RAD51 protein as an inhibitor of DNA re-replication. Besides its role in DSB repair by homologous recombination (HR; reviewed by Wright et al, 2018), RAD51 promotes fork reversal in response to replication stress (RS) and protects newly synthesized DNA from nucleolytic degradation (reviewed by Pasero and Vindigni, 2017; Bhat and Cortez, 2018; Berti et al, 2020). The latter function is independent of HR and involves the binding of RAD51 to dsDNA (Mason et al, 2019; Halder et al, 2022a). We show that upon origin re-firing, RAD51 molecules that are already bound to newly replicated DNA hinder the progression of re-replication forks, inducing frequent events of fork reversal and discontinuous DNA synthesis mediated by PRIMPOL primase. The ssDNA gaps generated by this mode of synthesis serve as entry points for MRE11 exonuclease, which further restricts over-replication through the degradation of re-duplicated DNA.

## Results

### RAD51 and FBH1 affect DNA re-replication

A genetic screening was designed to identify factors that influence the extent of re-replication caused by deregulated origin licensing. A doxycycline (dox)-inducible shRNA targeting GMN was integrated in HCT116, a colorectal cancer cell line that displays little chromosomal instability, maintains the DNA damage checkpoint and is sensitive to DNA re-replication (Ribas et al, 2003; Zhu et al, 2004). A library of endonuclease-prepared siRNA (esiRNA) molecules was designed against a selection of genes implicated in the cellular responses to DNA damage and RS, e.g., DNA helicases, DNA damage sensors, and chromatin remodeling factors (Table EV1). HCT116-shGMN cells treated with dox for 72 h were transfected with each component of the esiRNA library. Because DNA re-replication enlarges cell nuclei (Zhu et al, 2004), nuclear size was evaluated by high-throughput microscopy (HTM) 48 h after transfection. As expected, GMN downregulation markedly increased nuclear size due to over-replication (Fig. 1A,B, compare control and shGMN). Amongst the genes whose down-regulation enhanced or restricted this phenotype, RAD51 and FBH1 displayed strong but opposite effects (Fig. 1A). RAD51 mediates HR and protects the integrity of nascent DNA, whereas FBH1 displaces RAD51 from chromatin in response to RS and DNA damage (Fugger et al, 2009; Simandlova et al, 2013; Ronson et al, 2018). Consistent with their antagonistic functions, loss of RAD51 increased nuclear size, whereas loss of FBH1 reduced it. These effects were validated with independent siRNA molecules (Fig. 1C).

Beyond changes in nuclear size, the extent of actual DNA over-replication was monitored by flow cytometry analyses of BrdU incorporation and DNA content. Silencing of RAD51 in dox-treated HCT116-shGMN cells increased the percentage of cells with >4 C DNA content (from here on, "re-replicating cells"). In contrast, silencing of FBH1 decreased it (Fig. 1D,E). Because two siRNA molecules were combined to downregulate each gene, the effect of individual siRNAs was confirmed (Fig. EV1A,B). To test the role of RAD51 in DNA re-replication in a different cellular context, we used MLN4924, a drug that increases the levels of CDT1 protein leading to origin re-activation (Fu et al, 2021). The percentage of MLN4924-treated HCT116 cells undergoing re-replication was increased by RAD51 downregulation (Fig. EV1C). Restriction of re-replication by RAD51 was also observed in U2OS cells (Fig. EV1D). We noticed that loss of RAD51 by itself led to a slight increase in the levels of re-replication in both cell lines (Figs. 1D and EV1B,D).

We next evaluated how RAD51 or FBH1 depletion affected genomic integrity in the context of DNA re-replication. Loss of GMN induced DSBs that could be detected in neutral comet assays. FBH1 downregulation reduced the frequency of DSBs, while loss of RAD51 enhanced DNA re-replication without inducing more breaks than GMN depletion alone (Fig. 1F). GMN-depleted cells displayed phosphorylation of RPA32, CHK1 and H2AX. Moreover, the activation of p53 and the lack of H3-pSer10 suggested that cells did not enter mitosis (Fig. EV1E, lanes 1–2). Co-downregulation of FBH1 attenuated checkpoint activation and restored normal levels of H3 phosphorylation (Fig. EV1E, lanes 2 and 4). Consistent with the results of the comet assay, checkpoint proteins were not further activated by RAD51 co-depletion (Fig. EV1E, lanes 2 and 6). As expected, re-replication increased the percentage of apoptotic and dead cells (Fig. 1G,H). FBH1 downregulation restored the viability of GMN-depleted cells to control levels, while RAD51 co-depletion further reduced cell survival (Fig. 1G,H). Combined, these results demonstrate that RAD51 and FBH1 affect the extent of DNA re-replication when origin re-licensing regulation fails.

### RAD51 restricts DNA re-replication independently of HR

RAD51 has been involved in the repair of DNA damage induced by re-duplication (Truong et al, 2014). Immunofluorescence (IF) assays revealed that the amount of RAD51 on chromatin increased in parallel to the extent of re-replication and DNA damage (Figs. 2A,B and EV2A,B). At the peak of DNA over-replication (72 h post-shGMN), >20% of the cells were double-positive for chromatin-bound RAD51 and γH2AX (or RAD51 and RPA; Fig. EV2C,D). These analyses likely reflect the recruitment of RAD51 to repair DNA breaks generated in the context of re-replication.

To examine whether HR-mediated repair was required to limit re-replication extension, we tested the involvement of RAD51AP1, a protein that facilitates RAD51 recombinase activity (Wiese et al, 2007). Downregulation of RAD51AP1 did not change the percentage of cells undergoing re-replication upon GMN loss (Fig. 2C). Similarly, the percentage of re-replicating cells was not affected by RAD51 inhibitor B02 (Huang et al, 2011; Fig. 2D). As controls for RAD51AP1 siRNA efficiency and B02 activity, we confirmed the reported RPA phosphorylation in camptothecin-treated, RAD51AP1-depleted cells (Fig. EV2E; Parplys et al, 2015) and the accumulation of p21 in B02- and doxorubicin-treated cells (Fig. EV2F; Schürmann et al, 2021). These results suggest that

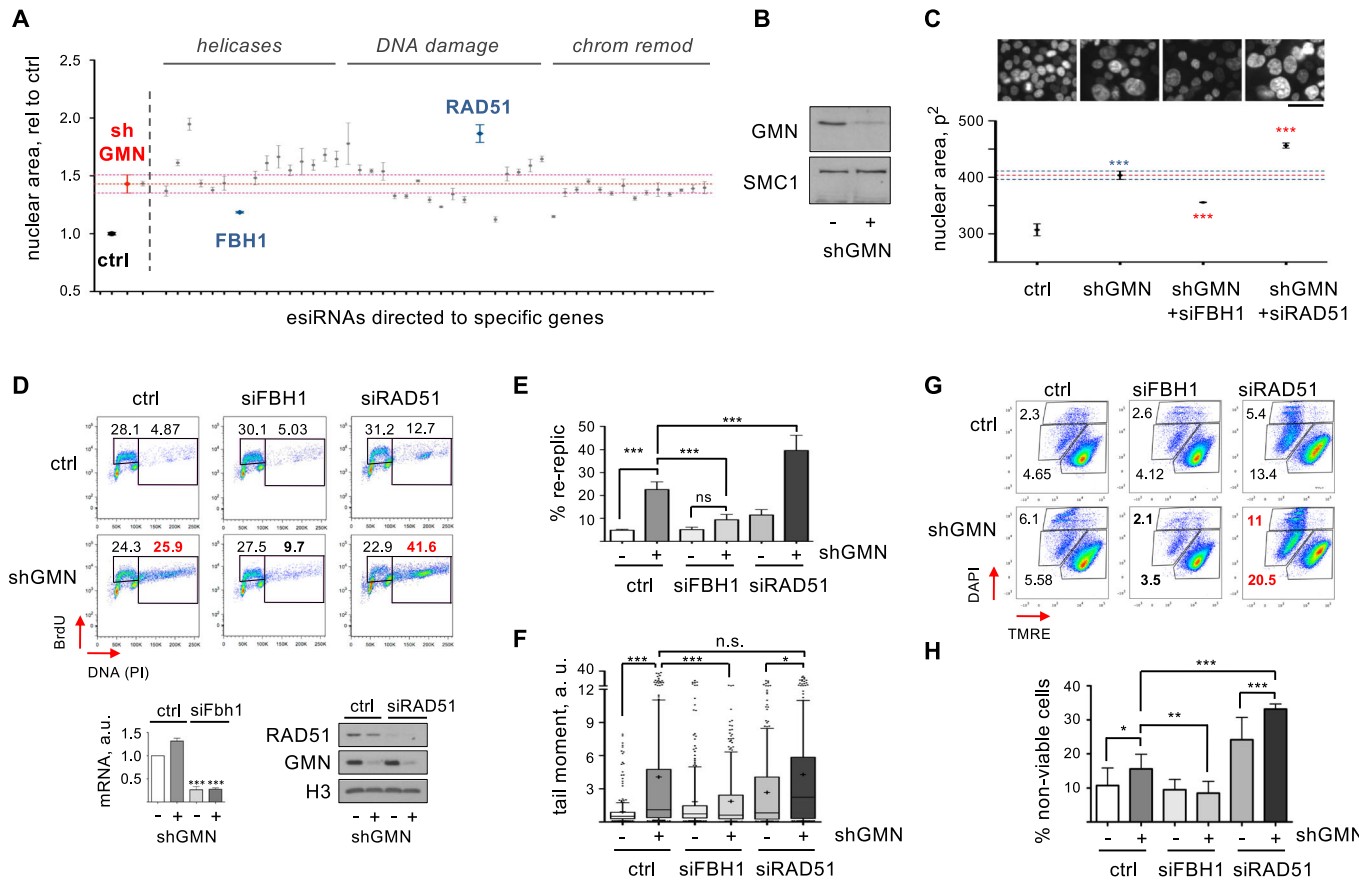

**Figure 1. RAD51 and FBH1 regulate DNA re-replication after GMN downregulation.**

(A) Nuclear size (area) of dox-treated HCT116-shGMN cells transfected with individual esiRNAs, normalized to control. Mean and SD of the median values from 3 replicates are represented. $n > 500$ cells per condition and replicate. (B) GMN levels in HCT116-shGMN cells before and after incubation with dox for 72 h. SMC1 is shown as loading control. One of three replicates is shown. (C) Top, DAPI-stained HCT116-shGMN cells treated with dox for 72 h (except control) and transfected with the indicated siRNAs for the last 48 h. Bar, 50 µm. Bottom, nuclear area (mean and SD) of the indicated samples. Three replicates were performed ($n > 900$ cells per condition and replicate). ***$p < 0.001$ (one-way Anova and Bonferroni's post-test). (D) Top, analysis of DNA over-replication in HCT116-shGMN cells grown with or without shGMN and transfected with the indicated siRNAs for 72 h. In each plot, the small gate indicates S phase cells with DNA content between 2 and 4 C and the large gate indicates cells with >4 C DNA. Bottom left histogram shows fold-change (mean and SD) of Fbh1 mRNA levels relative to control. $n = 3$ replicates. ***$p < 0.001$ (one-way Anova and Bonferroni's post-test). Bottom right, immunoblot detection of RAD51 and GMN proteins. H3 is shown as loading control. (E) Percentage (mean and SD) of cells undergoing re-replication. $n = 3$ replicates. ***$p < 0.001$; n.s., not significant (one-way Anova and Bonferroni's post-test). (F) Neutral comet assay in HCT116-shGMN cells grown with or without shGMN and transfected with the indicated siRNAs for 72 h. Box and whiskers plot represents tail moment. Boxes are drawn from the 25th to the 75th percentile. The central horizontal line indicates the median value. Whiskers are drawn from the 10th to the 90th percentile and the rest of values are drawn as individual dots (mimima, 0th percentile, maxima 100th percentile). Data from two different replicates are pooled. $n > 95$ cells per condition and replicate. ***$p < 0.001$; *$p < 0.05$; n.s., not significant (one-way Anova Kruskal–Wallis test and Dunn's post-test). (G) Cell viability in HCT116-shGMN cells grown with or without shGMN and transfected with the indicated siRNAs for 72 h. TMRE-negative staining (X axis) detects apoptotic cells. Positive DAPI staining (Y axis) detects dead cells. (H) Percentage (mean and SD) of non-viable cells (apoptotic + dead). $n = 3$ replicates. ***$p < 0.001$; **$p < 0.01$; *$p < 0.05$ (one-way Anova and Bonferroni's post-test). Source data are available online for this figure.

RAD51-dependent HR is not required for the control of DNA over-replication caused by deregulated origin licensing.

At least in yeast, HR-mediated repair of DSBs at forks stalled by R-loops may also lead to re-replication (Costantino and Koshland, 2018). In human cells, replication intermediates caused by origin reactivation may be cleaved by MUS81 endonuclease (Galanos et al, 2016), opening the possibility that their repair could extend re-replication. However, this effect is likely minimal in our cellular system because MUS81 downregulation, which promoted a detectable accumulation of control cells in G1 (Fig. EV2G; Naim et al, 2013), did not affect the percentage of re-replicating cells upon loss of GMN (Fig. 2E).

## RAD51 binding to chromatin modulates DNA re-replication

RAD51 has functions during DNA replication that do not require HR (reviewed by Bhat and Cortez, 2018; Berti et al, 2020). We considered that the role of RAD51 in preventing DNA re-replication could depend on its association to newly synthesized DNA in S phase. In this regard, higher amounts of chromatin-bound RAD51 were detected by IF in PCNA-positive than in PCNA-negative HCT116-shGMN control cells (Fig. 3A). Following biochemical fractionation of untreated cell populations sorted by DNA content, higher amounts of RAD51 were found in the chromatin fraction of S

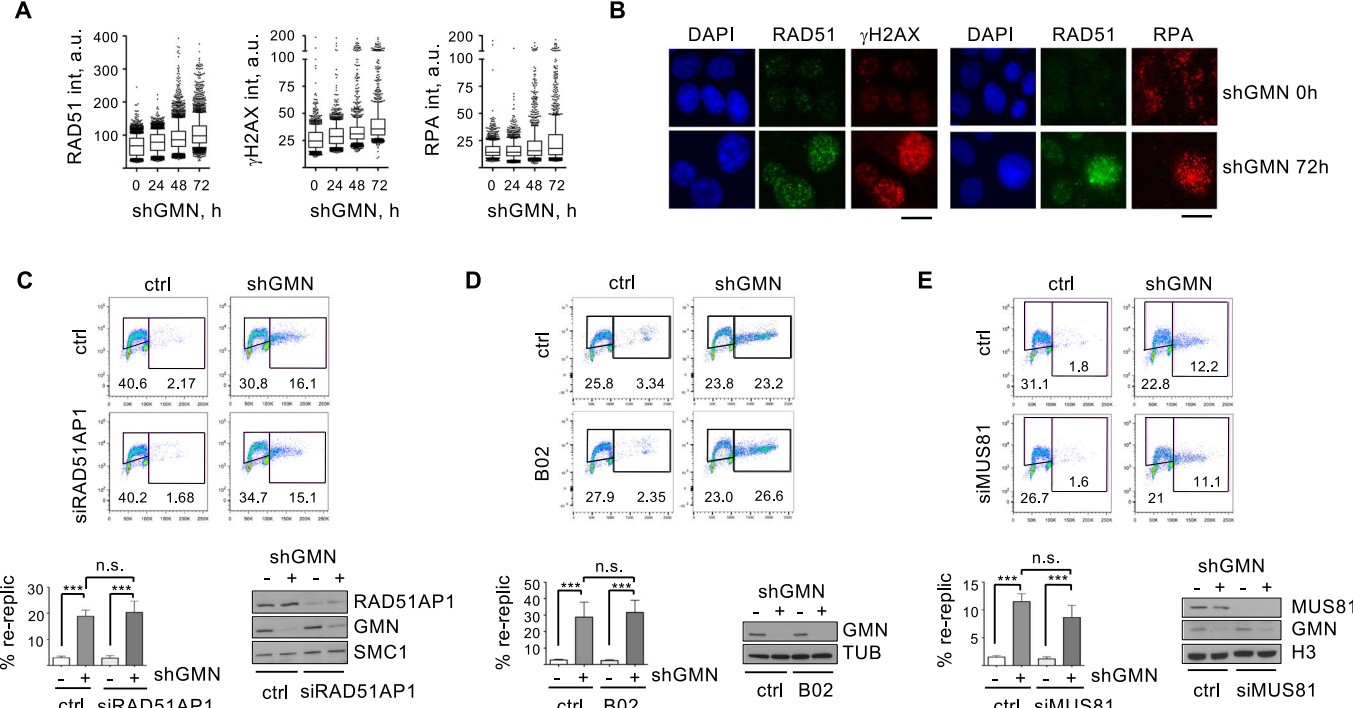

**Figure 2. HR factors and MUS81 do not regulate DNA re-replication.**

(A) Box and whiskers plot showing the distribution of RAD51, γH2AX, and RPA intensities in HCT116-shGMN cells at the indicated times after shGMN expression. Boxes are drawn from the 25th to the 75th percentile. The central horizontal line indicates the median value. Whiskers are drawn from the 10th to the 90th percentile and the rest of values are drawn as individual dots (mimima, 0th percentile, maxima 100th percentile). Data from 6 (RAD51) or 3 (γH2AX and RPA) replicates were pooled. $n = 400$ cells per condition and replicate. (B) Representative IF images of HCT116-shGMN cells with the indicated treatment and stained for chromatin-bound RAD51 and γH2AX (left), or chromatin-bound RPA (right) proteins. DNA was counterstained with DAPI. Bar, 15 μm. (C) Top, re-replication analysis in HCT116-shGMN cells grown with or without shGMN and transfected with RAD51AP1 siRNA for 72 h. Histogram shows the percentage (mean and SD) of cells undergoing re-replication. $n = 3$ replicates. ***$p < 0.001$; n.s., not significant (one-way Anova and Bonferroni's post-test). Bottom right, immunoblot detection of RAD51AP1 and GMN proteins. SMC1 is shown as loading control. (D) Same as (C) in HCT116-shGMN cells treated with 10 μM B02 for 24 h. $n = 4$ replicates. ***$p < 0.001$; n.s., not significant (one-way Anova and Bonferroni's post-test). Immunoblot shows GMN levels and TUBULIN (TUB) as a loading control. (E) Same as (C) and (D), in HCT116-shGMN cells transfected with MUS81 siRNA for 72 h. $n = 3$ replicates. ***$p < 0.001$; n.s., not significant (one-way Anova and Bonferroni's post-test). Immunoblots show the levels of MUS81 and GMN proteins. H3 is shown as loading control. Source data are available online for this figure.

and G2 phases relative to G1 (Fig. 3B). The inhibition of re-replication in the event of origin re-firing suggests that at least a fraction of chromatin-bound RAD51 could be located close to replication origins. Indeed, RAD51 ChIP-seq data (GEO:GSE91838) revealed an accumulation around ORC2-binding sites (Miotto et al, 2016), using randomized genomic positions as control (Fig. EV3A). RAD51 was also enriched around H2AZ-binding sites that correlate with origin activation (Long et al, 2020; Fig. EV3B). The enrichment of RAD51 around active origins was confirmed using short nascent strand sequencing (SNS-seq) data. The enrichment of RAD51 around SNS-seq positions (Picard et al, 2014; Fig. EV3C) was particularly clear at those origins identified by both ORC2 and SNS-seq signals (Fig. EV3D).

We hypothesized that the binding of RAD51 protein to newly synthesized DNA in S phase is responsible for the restriction of re-replication. In support of this idea, downregulation of FBH1, which prevented the extent of re-replication upon GMN loss (Fig. 1D), stabilized RAD51 on chromatin (Fugger et al, 2009; Simandlova et al, 2013). We have confirmed this effect by IF using two separate RAD51 antibodies (Figs. 3C and EV4A). RAD51 stabilization on chromatin was also observed after downregulation of RAD54 translocase, which

destabilizes RAD51 filaments on undamaged dsDNA (Solinger et al, 2002; Shah et al, 2010; Mason et al, 2015; Figs. 3C and EV4B). Accordingly, RAD54 downregulation restricted re-replication upon GMN depletion (Fig. 3D). The correlation between the presence of RAD51 on undamaged DNA and the extent of re-replication was also confirmed with a non-genetic approach. RS-1, a small molecule that stabilizes RAD51 binding to DNA (Jayathilaka et al, 2008; Mason et al, 2014) increased RAD51 IF signal (Fig. EV4C) and decreased re-replication levels (Fig. 3E), even if the effect was milder than FBH1 or RAD54 ablation. In turn, ectopic expression of RAD51^K133A, a dominant-negative ATP-binding mutant that interferes with the formation of RAD51 filaments (Chi et al, 2006; Kim et al, 2012), had the opposite effect, increasing the extent of re-replication in HCT116-shGMN cells (Fig. EV4D). Combined, these results suggest that the stabilization of RAD51 protein on chromatin during S phase limits DNA re-replication after origin refiring.

Of note, the genetic manipulations that led to an accumulation of RAD51 on DNA and restricted re-replication may have different effects on DSB repair. For instance, an inactive FBH1 mutant results in hyper-recombination (Simandlova et al, 2013), while loss of RAD54 impairs HR-dependent repair (Heyer et al, 2006). In

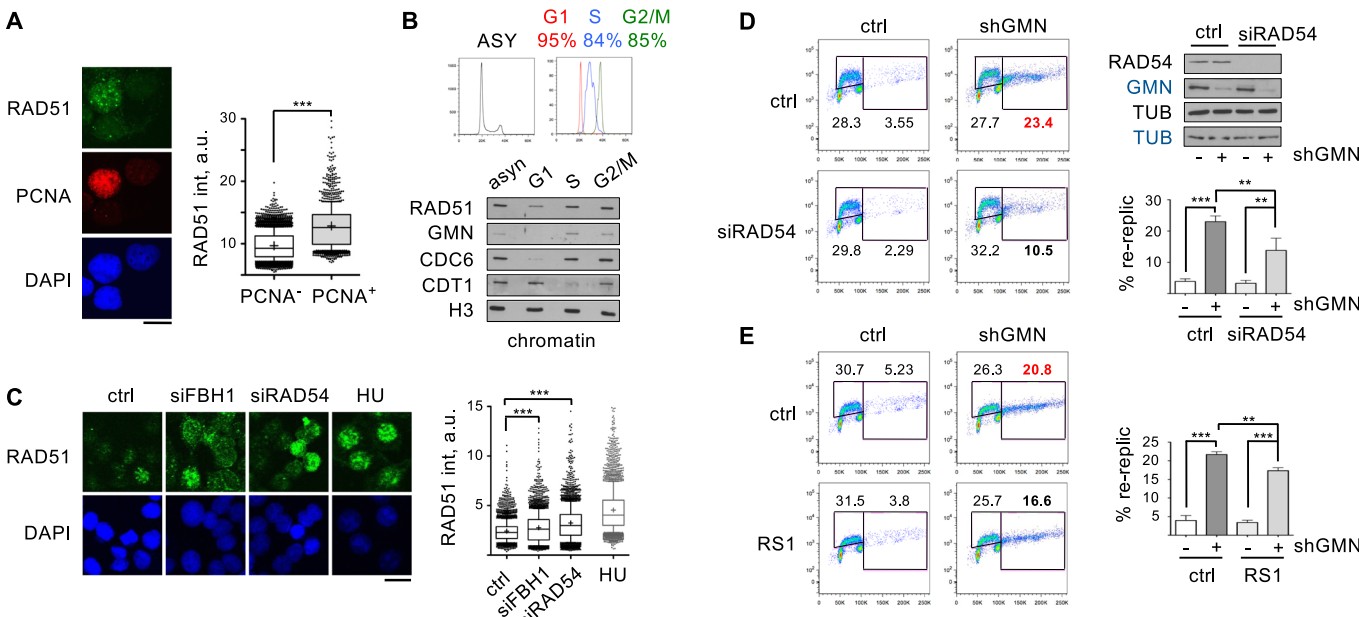

**Figure 3. Chromatin-bound RAD51 restricts DNA re-replication.**

(A) Left, representative IF images of asynchronous HCT116-shGMN cells stained for chromatin-bound RAD51 and PCNA. DNA was counterstained with DAPI. Bar, 15 µm. Right, box and whiskers plot showing the distribution of RAD51 intensity in PCNA-negative and PCNA-positive cells. Data from two replicates were pooled. $n = 4954$ (PCNA-negative) and 1523 (PCNA-positive) cells. ***$p < 0.0001$ (Mann–Whitney test). (B) Top, flow cytometry profiles of DNA content of asynchronous HCT116 cells (left), or cell populations sorted in the G1, S and G2/M phases of the cell cycle. Bottom immunoblots show the levels of the indicated proteins on chromatin. H3 is shown as loading control. (C) Representative IF images of chromatin-bound RAD51 protein in HCT116-shGMN cells transfected with the indicated siRNAs for 72 h, or treated with 3 mM HU for 24 h as a positive control. DNA was counterstained with DAPI. Bar, 20 µm. Right, box and whiskers plot showing the distribution of RAD51 intensity. Data from 3 experiments are pooled ($n = 1000$ cells per condition and replicate). ***$p < 0.0001$ (Kruskal–Wallis and Dunn's post-test). (D) Left, re-replication analysis in HCT116-shGMN cells (with or without shGMN) transfected when indicated with RAD54 siRNA for 72 h. Top right, immunoblot detection of RAD54 and GMN proteins. TUBULIN (TUB) is shown as loading control. Bottom right, percentage (mean and SD) of cells undergoing re-replication. $n = 3$ replicates. ***$p < 0.001$; **$p < 0.01$ (one-way Anova and Bonferroni's post-test). (E) Same as (D), in cells treated with 7.5 µM RS1 for 24 h. Histogram shows the percentage (mean and SD) of cells undergoing re-replication. $n = 3$ replicates. ***$p < 0.001$; **$p < 0.01$ (one-way Anova and Bonferroni's post-test). Box plots in (A, C): Boxes are drawn from the 25th to the 75th percentile. Median values are indicated by horizontal lines and mean values by the "+" symbol. Whiskers are drawn from the 10th to the 90th percentile and the rest of values are drawn as individual dots (mimima, 0th percentile, maxima 100th percentile). Source data are available online for this figure.

agreement with our previous observations, FBH1 downregulation reduced the extent of γH2AX and chromatin-bound RAD51 in GMN-depleted cells, indicative of proper DNA repair (Fig. EV4E). Conversely, γH2AX and RAD51 signals remained high following RAD54 depletion (Fig. EV4E), suggesting that loss of RAD54 restricted the extent of DNA re-replication and prevented HR-dependent repair of DSBs.

## RAD51 blocks the progression of re-replication forks

At the mechanistic level, RAD51 might restrict over-replication by preventing origin re-activation or by slowing down re-replication forks. These possibilities were tested in stretched DNA fiber assays (Fig. 4A). Previous work has shown that DNA re-replication induced by accumulation of CDT1 reduces the speed of fork progression rate (FR; Muñoz et al, 2017; Fu et al, 2021). As expected, GMN downregulation recapitulated this effect (Fig. 4B). Because FR and origin activity are frequently linked (Rodríguez-Acebes et al, 2018), GMN-depleted cells also displayed a slight increase in origin firing (Fig. 4C). Downregulation of RAD51 affected the percentage of active origins only marginally, arguing against a role in preventing origin re-firing (Fig. 4C). While loss of RAD51 had virtually no effect on FR in the presence of GMN, forks

became much faster when RAD51 was downregulated in cells lacking GMN (Fig. 4B).

RAD51 mediates fork reversal in response to different challenges, causing a global decrease in FR (Henry-Mowatt et al, 2003; Ray Chaudhuri et al, 2012; Zellweger et al, 2015). Upon RS, many forks are reversed in an ATR-dependent response regardless of their proximity to a replication block (Mutreja et al, 2018). Because fork reversal has also been observed after origin deregulation (Neelsen et al, 2013), we investigated its participation in the phenotypes caused by GMN and RAD51 downregulation. Reversed forks were visualized by electron microscopy in re-replicating cells. To rule out effects of GMN downregulation on fork reversal that could be independent of re-replication, the control condition were HCT116-shGMN cells after 24 h of shGMN expression, when GMN levels were reduced but no detectable re-replication had taken place (Fig. EV2B). As anticipated, the percentage of reversed forks was markedly increased after 72 h of GMN downregulation, and this effect was strictly dependent on the presence of RAD51 (Fig. 4D).

The link between fork reversal and FR in re-replicating cells was further tested using PARP1 inhibitor olaparib, which restricts fork reversal and increases FR in response to RS and DNA damage (Ray Chaudhuri et al, 2012; Berti et al, 2013). Olaparib increased FR in cells lacking GMN (Fig. 4E), but to a lower extent than RAD51

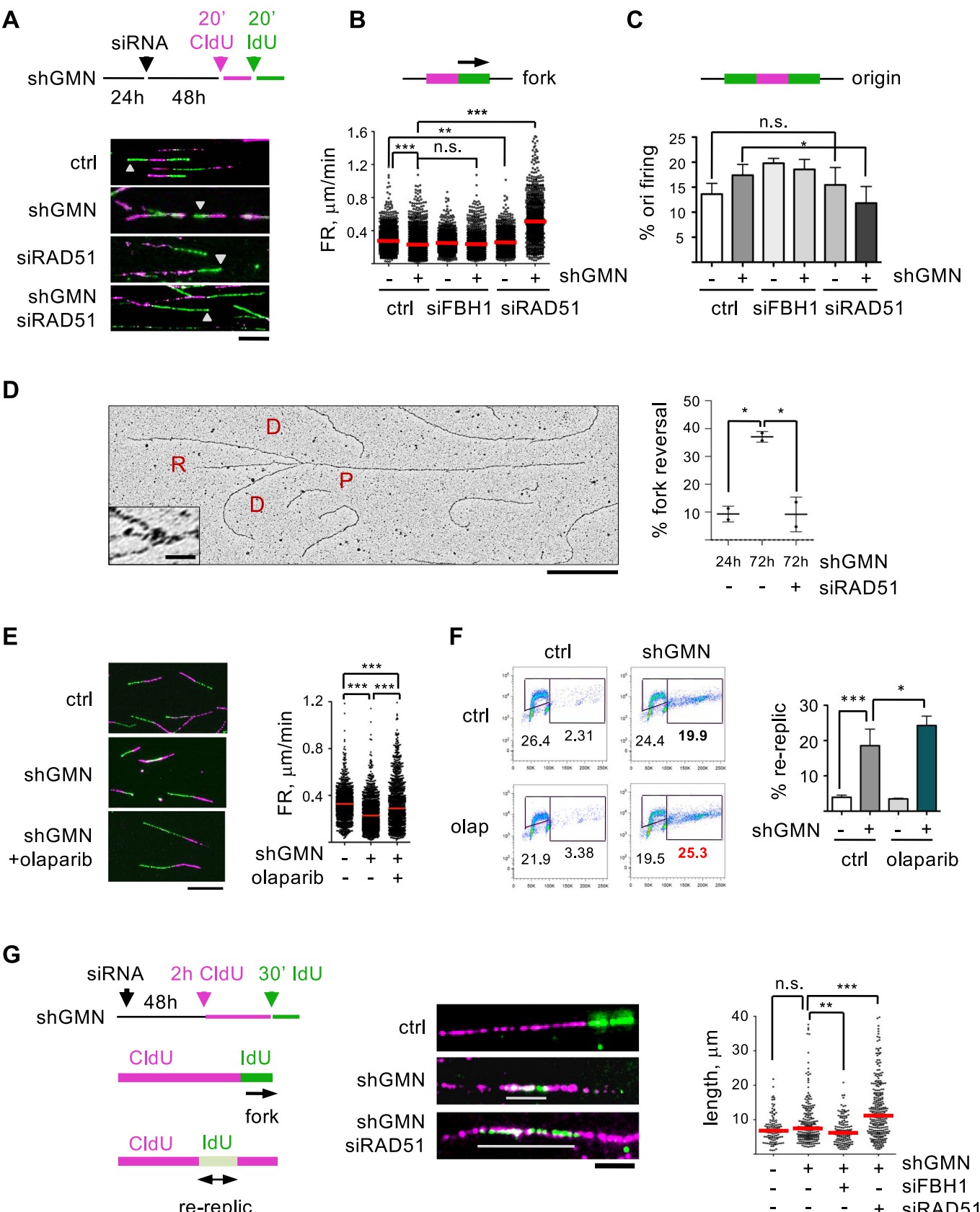

**Figure 4. RAD51 blocks re-replication fork progression.**

(A) Experimental schematic and examples of DNA fibers derived from HCT116-shGMN cells. CldU- and IdU-containing tracks are visualized in magenta and green, respectively. Gray arrowheads indicate forks. Bar, 10 µm. (B) Schematic of a labeled active fork and distribution of FR values in the indicated samples. Red bars, median values. Data from 5 (control −/+ shGMN) and 3 siRNA-treated replicates are pooled. $n > 297$ FR measurements per condition and assay. ***$p < 0.001$; **$p < 0.01$; n.s., not significant (Kruskal–Wallis and Dunn's post-test). (C) Schematic of a labeled replication origin and percentage (mean and SD) of structures corresponding to origin firing in the indicated samples. $n = 5$ replicates for control (−/+ shGMN) and 3 for siRNA-treated cells. $n > 500$ replicative structures per condition and assay. *$p < 0.05$; n.s., not significant (one-way Anova and Bonferroni's post-test). (D) Detection of reversed forks in HCT116-shGMN cells in the presence of shGMN for 24 h and transfected with RAD51 siRNA when indicated. A representative EM micrograph is shown. P, parental strand; D, daughter strands; R, regressed arm. Bar, 500 nm (inset bar, 20 nm). Histogram shows the percentage (mean and SD) of reversed forks in each condition. $n = 2$ replicates (132, 140, and 133 total forks scored in each condition). *$p < 0.05$ (one-way Anova and Bonferroni's post-test). (E) Examples of DNA fibers in HCT116-shGMN cells in the indicated conditions. Bar, 15 µm. When indicated, cells were pre-treated with olaparib (10 µM, 2 h) and sequentially pulsed with CldU and IdU for 25 min. Right, distribution of FR values. Red bars, median values. Data from 3 replicates are pooled. $n > 300$ FR measurements per condition and replicate. ***$p < 0.001$ (Kruskal–Wallis and Dunn's post-test). (F) Analysis of re-replication in HCT116-shGMN cells grown with or without shGMN (72 h) and olaparib (1 µM for the last 24 h). Histogram shows the percentage (mean and SD) of cells undergoing re-replication. $n = 3$ replicates. ***$p < 0.001$; *$p < 0.05$ (one-way Anova and Bonferroni's post-test). (G) Experimental design to detect re-replication events. Re-replicated DNA tracks are white (overlap of magenta and green) and indicated with gray lines. Bar, 5 µm. Right graph: length (median and distribution) of re-replicated tracks in the indicated samples. Data from three independent experiments are pooled. Total number of re-replicated tracks/ total number of scored structures were as follows: 111/1602 (control); 234/1557 (shGMN); 139/1551 (shGMN+siFBH1); 280/1551 (shGMN+siRAD51). ***$p < 0.001$; **$p < 0.01$; n.s., not significant (Kruskal–Wallis and Dunn's post-test). Source data are available online for this figure.

downregulation (compare Fig. 4B,E). Besides its effect on FR, olaparib elevated the percentage of cells undergoing re-replication upon GMN loss (Fig. 4F), indicating that fork reversal also contributes to limit DNA over-replication. In this regard, down-regulation of SMARCAL1 translocase, but not HLTF, slightly increased re-replication (Fig. EV5A,B).

Inhibition of fork reversal had a milder effect in preventing re-replication than RAD51 downregulation, suggesting an additional function for the latter. We considered that RAD51 protein might hinder directly the progression of re-replication forks. To test this possibility, we used a variation of DNA fiber assay that increases the chances of capturing origin re-firing events (Neelsen et al, 2013; Muñoz et al, 2017). A longer first pulse with CldU (magenta) was followed by a shorter IdU pulse (green). Origins that are activated twice are visualized as white stretches (merge of both colors) embedded within magenta structures (Fig. 4G). The percentage of re-replicated tracks upon GMN loss was not affected by co-depletion of RAD51 (Fig. EV5C), confirming that RAD51 does not regulate origin re-firing events. However, re-replicated tracks were much longer in the absence of RAD51, while FBH1 loss had the opposite effect (Figs. 4G and EV5D). These results support the direct participation of chromatin-bound RAD51 in hindering the progression of re-replication forks.

## Discontinuous DNA synthesis at re-replication forks

Deregulation of origin activity correlates with the accumulation of ssDNA gaps (Neelsen et al, 2013). To test the effect of RAD51 in this process, parental DNA was continuously labeled with BrdU for 24 h, followed by incubation in fresh medium and BrdU IF visualization in native conditions, which only mark its presence at ssDNA gaps. As expected, GMN downregulation increased the percentage of cells positive for native BrdU foci (Fig. 5A). Surprisingly, co-depletion of RAD51 restricted this effect (Fig. 5A), suggesting that RAD51 mediates ssDNA gap formation during re-replication. This observation was confirmed by IF detection of RPA on chromatin: GMN downregulation increased the number of RPA foci per cell, whereas co-downregulation of RAD51 reduced it (Fig. 5B). Upon GMN loss, RPA-positive cells displayed larger nuclei, consistent with higher re-replication (Fig. EV6A). The positive correlation between total DNA content and number of RPA foci in cells undergoing re-replication was attenuated by

RAD51 downregulation (Fig. EV6B). Therefore, the combined downregulation of GMN and RAD51 yields cells with higher DNA content due to re-replication, but fewer RPA foci. These results suggest a discontinuous mode of DNA synthesis at re-replication forks that depends on the presence of RAD51.

## Progression of re-replication forks depends on PRIMPOL and MRE11

The discontinuous DNA synthesis observed during the second round of replication would require events of repriming of DNA synthesis, suggesting the participation of specialized primase PRIMPOL (Mourón et al, 2013; Piberger et al, 2020; Quinet et al, 2020; González-Acosta et al, 2021). As anticipated, PRIMPOL downregulation reduced the percentage of GMN-depleted cells undergoing re-replication (Fig. 5C). If RAD51 blocks re-replication forks and PRIMPOL promotes their reactivation, we reasoned that in the absence of RAD51, forks would not be paused and PRIMPOL should be dispensable. Indeed, the inhibitory effect caused by PRIMPOL silencing on re-replicating cells was almost abolished when RAD51 was downregulated (Fig. 5D). Other fork restart mechanisms could facilitate re-replication, such as XRCC3, a RAD51 paralog that promotes re-initiation of DNA synthesis but is not involved in fork reversal (Henry-Mowatt et al, 2003; Petermann et al, 2010; Berti et al, 2020). To a lesser extent than PRIMPOL loss, XRCC3 downregulation also reduced re-replication (Fig. EV6C).

The reversed forks and ssDNA gaps generated during re-replication could be substrates for MRE11 nuclease (Hashimoto et al, 2010; Schlacher et al, 2011; Mijic et al, 2017; Lemaçon et al, 2017). Consistent with this notion, chemical inhibition of MRE11 for 24 h enhanced the percentage of cells with re-replicated DNA, compared to untreated GMN-depleted cells (Fig. 5E). HCT116 cells carry a splicing mutation in the MRE11 gene (Giannini et al, 2002) and express lower levels of MRE11 protein than other cancer cell lines. Still, MRE11 protein levels were readily detected by immunoblot and not affected by mirin (Figs. 5E and EV6D). When MRE11 inhibition was combined with RAD51 down-regulation, the percentage of cells with over-replicated DNA was further increased, probably reflecting the fact that RAD51 protects nascent DNA from exonucleolytic degradation (Fig. EV6E). Together, these results indicate that DNA re-replication in the

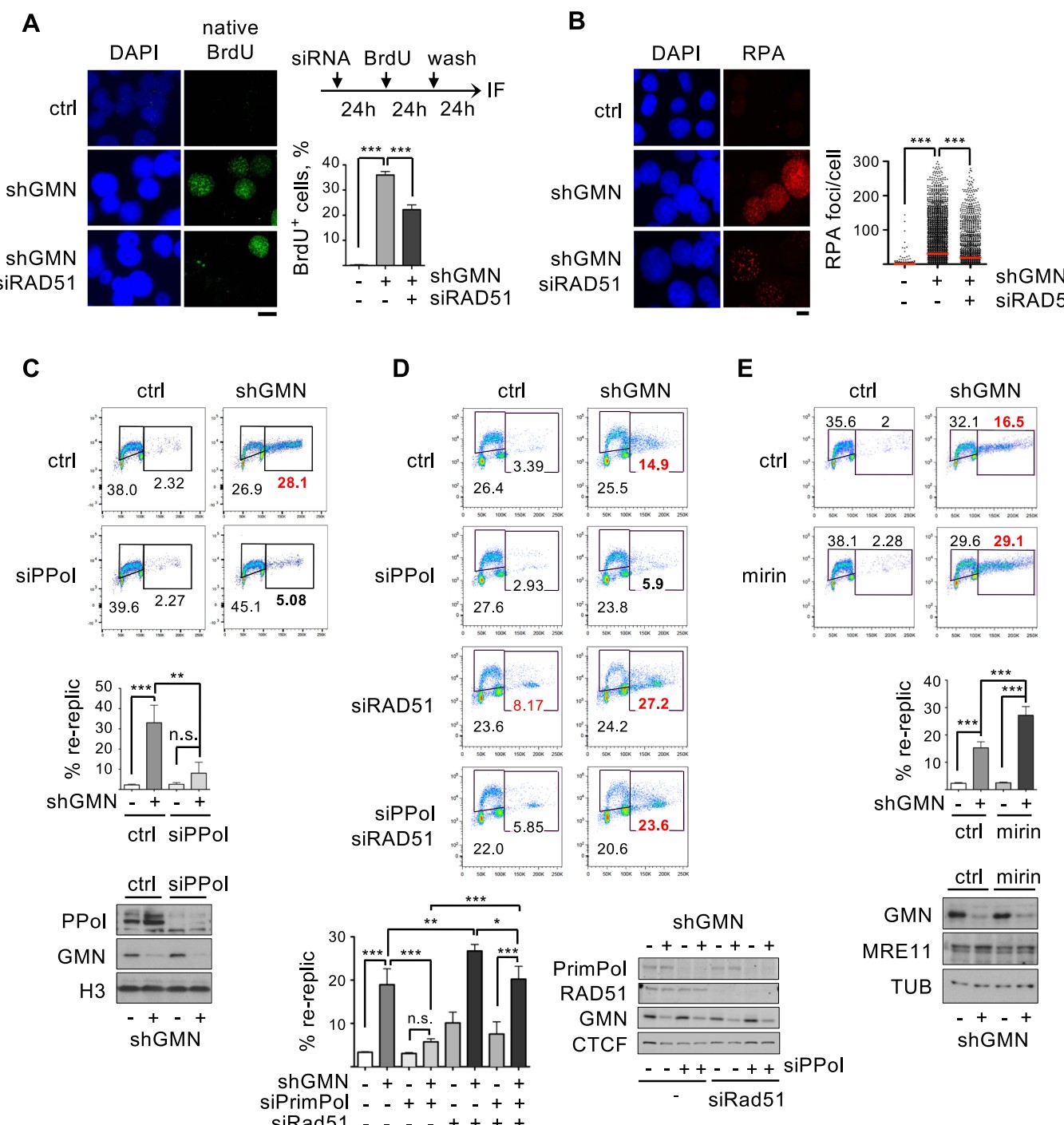

**Figure 5. Discontinuous DNA synthesis during re-replication.**

(A) Left, images of BrdU detection by IF in native conditions in HCT116-shGMN cells grown with or without shGMN and transfected with RAD51 siRNA for 72 h when indicated. DNA was counterstained with DAPI. Bar, 10 μm. Right, experimental schematic and percentage (mean and SD) of cells positive for native BrdU staining (>10 foci). $n = 3$ replicates. $n > 500$ cells scored per condition and replicate. ***$p < 0.001$ (one-way Anova and Bonferroni's post-test). (B) Left, images of RPA detection in HCT116-shGMN cells grown as in (A). Bar, 10 μm. Right, number of RPA foci per cell (mean and SD), pooled from 4 experimental replicates. $n > 3300$ cells scored per condition. ***$p < 0.001$ (Kruskal–Wallis and Dunn's post-test). (C) Top, re-replication analysis in HCT116-shGMN cells grown with or without shGMN and transfected with PRIMPOL siRNA (siPPol) for 72 h when indicated. Bottom, percentage (mean and SD) of cells undergoing re-replication. $n = 3$ replicates. ***$p < 0.001$; **$p < 0.01$; n.s., not significative (one-way Anova and Bonferroni's post-test). Immunoblots show levels of PRIMPOL and GMN. H3 is shown as loading control. (D) Same as in (C) in HCT116-shGMN cells transfected with RAD51 and/or PRIMPOL siRNAs for 72 h as indicated. Bottom, percentage (mean and SD) of cells undergoing re-replication. $n = 3$. ***$p < 0.001$; **$p < 0.01$; *$p < 0.05$; n.s., not significant (one-way Anova and Bonferroni's post-test). Immunoblots show the levels of RAD51, PrimPol, and GMN. CTCF is shown as loading control. (E) Same as (C), in HCT116-shGMN cells treated with or without shGMN (72 h) and mirin (10 μM, last 24 h) as indicated. Bottom, percentage (mean and SD) of cells undergoing re-replication. $n = 3$. ***$p < 0.001$ (one-way Anova and Bonferroni's post-test). Immunoblots show levels of MRE11 and GMN. TUBULIN (TUB) is shown as loading control. Source data are available online for this figure.

presence of chromatin-bound RAD51 enhances fork reversal and discontinuous DNA synthesis mediated by PRIMPOL. The ssDNA gaps left by PRIMPOL may facilitate MRE11 access to re-duplicated DNA, contributing to its elimination.

## Discussion

We have unveiled an unanticipated role of RAD51 in hindering the progression of forks derived from re-fired replication origins. To our knowledge, this is the first evidence for a layer of protection against re-replication when the controls over origin re-licensing have failed. Previous genetic screenings had identified genes that induce DNA re-replication and/or endoreduplication de novo (Vassilev et al, 2016), while our screening has addressed the relevance of a group of genes in modulating the extent of re-replication caused by loss of GMN function. With this approach, we found that RAD51 strongly restricts DNA re-replication. Loss of RAD51 also induced a modest amount of re-replication in the presence of GMN, suggesting that origin re-firing events are more frequent than previously presumed. We propose that RAD51 prevents genomic duplications that could arise from re-fired origins under physiological circumstances, and this function becomes critical when origin re-firing is exacerbated by chemical or genetic alterations.

Our model for this novel function of RAD51 is summarized in Fig. 6. After the first round of origin firing, RAD51 is temporarily bound to newly synthesized DNA, either nucleating from ssDNA gaps or directly on dsDNA (i–ii). In the event of origin re-firing, the presence of RAD51 on the template becomes a physical impediment to the progression or re-replicated forks (iii), slowing down their progression and promoting fork reversal (iv), which serves as a second brake for the extension of re-replication. Stalled and/or reversed forks are eventually reactivated by the action of PRIMPOL primase downstream of the pausing points, leading to discontinuous DNA synthesis and the accumulation of ssDNA gaps at re-duplicated DNA (v). Finally, MRE11 nuclease may contribute to restraining re-replication by degrading over-duplicated DNA (vi). Re-started forks may again encounter RAD51 proteins, triggering another cycle of blocking, reversal, and restart (iv–vi).

A central element in our model is the temporal association of RAD51 protein with newly synthesized DNA, which is supported by several lines of evidence. The amount of chromatin-bound RAD51 is higher in S-phase than in G1 (Fugger et al, 2009; Hashimoto et al, 2010; Somyajit et al, 2015; Zellweger et al, 2015; this study). We have observed an enrichment of RAD51 at or around origins identified by ORC2- and H2AZ-binding sites, as well as SNS positions (Fig. EV3A–C). RAD51 is detected both at replication forks and mature chromatin by iPOND (Sirbu et al, 2011; Kim et al, 2012; Zellweger et al, 2015; Bétous et al, 2018). In *X. laevis* egg extracts, RAD51 binds to DNA after origin activation and stays on chromatin after fork components have been evicted (Hashimoto et al, 2010). Through this association, RAD51 protects ssDNA gaps left behind the fork and at reversed structures (reviewed by Pasero and Vindigni, 2017).

The binding of RAD51 to newly synthesized DNA could be initiated at ssDNA gaps or directly at dsDNA. RAD51 has the capacity to bind to dsDNA both in vitro and in vivo (reviewed in Reitz et al, 2021) and is capable of forming nucleofilaments on replicated dsDNA (Shah et al, 2010; Mason et al, 2015). A recent

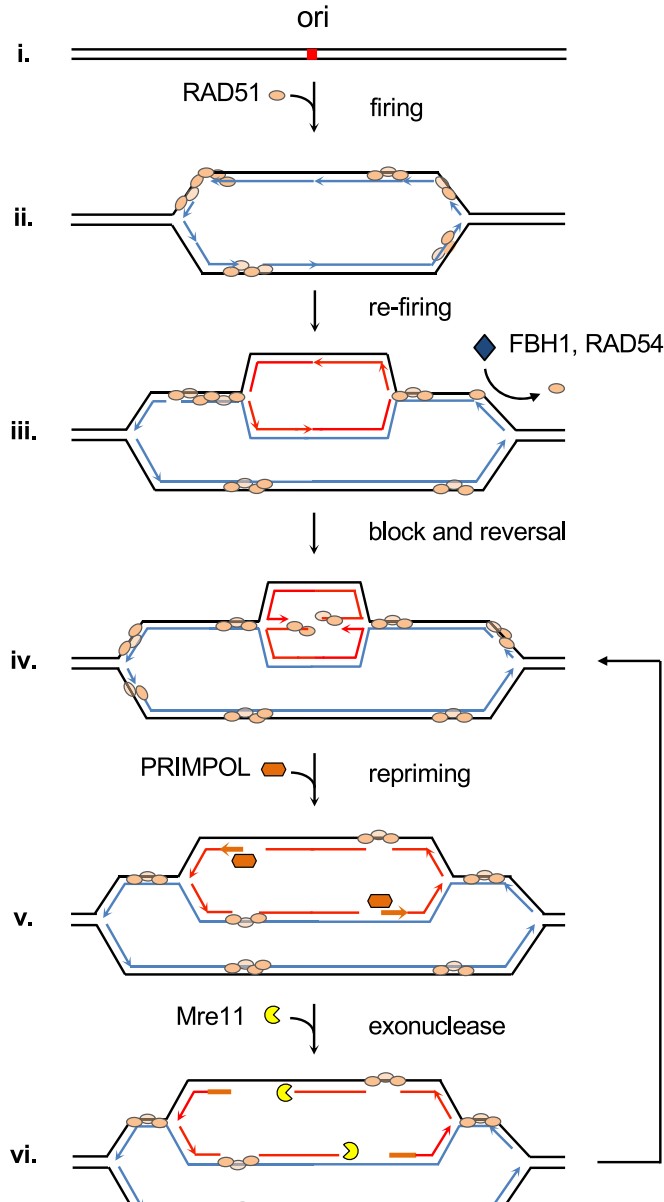

**Figure 6. RAD51-mediated restriction of DNA re-replication: a model.**

After origin firing, RAD51 protein associates with newly replicated chromatin (i–ii). In the event of origin refiring, re-replication fork progression is initially hindered by the presence of RAD51 in the new template DNA (iii), resulting in frequent fork reversal (iv). Anti-recombinases or proteins that displace RAD51 from the DNA (e.g., FBH1 and RAD54) may facilitate DNA re-duplication. PRIMPOL-mediated reactivation of stalled and reversed re-replication forks generates ssDNA gaps (v). The subsequent entry of MRE11 exonuclease contributes to the clearing of re-duplicated DNA (vi). The processes described between (iv) and (vi) could occur several times on the same molecule. Line colors: black, parental DNA; blue, newly synthesized DNA (1st round); red, newly synthesized DNA at re-replication forks; orange (thicker) lines: re-priming by PRIMPOL. See text for details.

study proposes that the protective function of RAD51 against nucleases involves binding to dsDNA (Halder et al, 2022a). In agreement with the model in Fig. 6, several genetic or chemical interventions that increased the amount of RAD51 on chromatin resulted in lower levels of re-replication after GMN loss.

The same RAD51 molecules that bind and protect newly replicated DNA are likely responsible for preventing the spread of re-replication forks. This function might explain the lower processivity observed in re-replication forks in different organisms (Green et al, 2006; Tanny et al, 2006; Finn and Li, 2013; Muñoz et al, 2017; Fu et al, 2021). Notably, in one of the few known examples of physiological DNA re-duplication, the progression of re-replication forks in *D. melanogaster* follicle cells was enhanced in the absence of RAD51 homologs spindle-A and spindle-B. This result has been interpreted as HR-mediated repair of DSBs slowing down re-replication forks (Alexander et al, 2016). In the light of our new results, it is possible that the direct inhibition of fork progression by RAD51 homologs might also contribute to the phenotype.

RAD51 downregulation reduced the percentage of reversed forks caused by GMN loss, underscoring the implication of fork reversal as an additional brake to re-replication. In fact, prevention of fork reversal with olaparib increased fork rate and the extent of re-replication in GMN-depleted cells. In the context of GMN loss, reversal of re-replication forks likely depends on the SMARCAL1, HLTF and ZRANB3 translocases pathway instead of FBH1 (Liu et al, 2020). While SMARCAL1 downregulation increased re-replication, we did not find evidence for the participation of HLTF, a fact that could reflect the different biochemical requirements and activities displayed by these translocases (Halder et al, 2022b).

DNA re-replication also relied on PRIMPOL primase, revealing a discontinuous mode of DNA synthesis that explains the high frequency of ssDNA gaps observed upon origin re-firing (Neelsen et al, 2013; this study). Downregulation of RAD51 virtually eliminated the need for PRIMPOL and the accumulation of ssDNA gaps, linking discontinuous DNA synthesis to the presence of RAD51 on DNA. Besides, PRIMPOL-induced gaps allow the access of MRE11 nuclease to re-replicated DNA. Besides MRE11, it is possible that other nucleases such as DNA2, EXO1, or CtIP (Pasero and Vindigni, 2017) contribute to the clearance of excess DNA. The generation of RS after origin re-firing and the involvement of PRIMPOL primase bears parallelisms with the situation of stem cells that undergo RS related to their short G1 phase (Ahuja et al, 2016). In this regard, we have recently reported that hematopoietic stem cells rely on PRIMPOL to sustain rapid replication when forced to proliferate (Jacobs et al, 2022).

We have considered the possibility that RAD51 could restrict break-induced replication (BIR), a mechanism that extends re-replication in vitro (Johansson and Diffley, 2021) and in vivo at yeast centromeres (Hanlon and Li, 2015). However, this effect is likely minimal in our system because downregulation of MUS81 and RAD51AP1, two factors involved in BIR-related processes (Minocherhomji et al, 2015; Yadav et al, 2022), barely affected re-replication levels.

The fact that HR participates in the repair of DSBs generated during re-replication (Truong et al, 2014; reviewed by Alexander and Orr-Weaver, 2016) is compatible with the HR-independent role of RAD51 in preventing the extension of re-replicated forks. We propose that, in the event of origin re-firing, RAD51 first limits the extension of re-replicated DNA, reducing the chance of potential genomic duplications. Then, if DSBs are generated in the process, its recombinase function would mediate post-replicative repair to maintain genome integrity. In support of these two complementary functions, FBH1 downregulation reduced re-replication without affecting HR repair of DSBs, while RAD54 silencing prevented the re-replication and impeded the repair of DNA damage.

DNA re-replication could fuel carcinogenesis by promoting aneuploidy and the formation of heterogeneous cell populations that enhance the adaptability of tumor cells (Rapsomaniki et al, 2021). Considering that modest levels of re-replication fail to trigger DNA damage checkpoints (Neelsen et al, 2013) and are tolerated in vivo (Muñoz et al, 2017), restricting the consequences of origin re-initiation may be particularly important in pre-tumoral lesions, when origin licensing proteins are frequently overexpressed and oncogenic signals interfere with origin regulation. In this context, RAD51 would limit the chances of acquiring oncogene amplifications and/or translocations at over-duplicated regions.

# Methods

## HCT116-shGMN cell line

Stable HCT116-shGMN cell lines were generated using a TRIPZ vector carrying an inducible shRNA molecule targeting the 3′ UTR of GMN gene (5′-TATGTAGTTATGTACTCTG-3′; clone ID: V2THS_114945; GE Healthcare, USA). Parental HCT116 cells (RRID:CVCL_0291) were infected with the lentiviral vector and selected with 5 µg/ml puromycin for 5 days. Resistant colonies were pooled and maintained in DMEM + 10% fetal bovine serum + 10% pen/strep + 0.5 µg/ml puromycin. Expression of GMN shRNA was induced with 2 µg/ml doxycycline (dox) for 3 days. U2OS cells (RRID: CVCL_0042) were obtained from the CNIO cell line repository. Cell cultures were tested periodically for mycoplasma contamination. Lipofectamine 2000 (Invitrogen) was used for plasmid and siRNA transfections.

## esiRNA-based screening

HCT-116 shGMN cells were cultured in µCLEAR bottom polylysine-treated 96-well plates (Greiner Bio-One) in triplicates in the presence of 2 µg/ml dox. 24 h after seeding, cells were transfected with individual esiRNA molecules (30 nM) (Sigma-Aldrich). Slx4 and Rad54 genes were targeted with siRNA instead of esiRNA. 72 h after seeding, cells were fixed with 2% PFA (10 min/ RT) and stained with 1 µg/ml DAPI (Sigma) for 3 min. Plates were analyzed in an Opera High-Content Screening System (PerkinElmer, USA) with an APO 20×, 0.7 NA water-immersion objective. Nuclear area was measured with Acapella software v2.6 (PerkinElmer, USA). The median nuclear size in each well was quantified, relative to the average of medians of control (-dox) cells.

## Plasmids, antibodies, and oligonucleotides

RAD51 WT and K133A mutant cDNAs were kindly provided by Dr. P. Sung (UT Health San Antonio, TX, USA) and cloned into Gateway pDEST-V5 or pDEST47 vectors (Thermofisher Scientific). Antibodies, siRNAs, oligonucleotides, and other reagents used in this study are listed in Tables EV2–EV5.

## RNA isolation, reverse transcription, and quantitative PCR (RT-qPCR)

Total RNA was isolated with Trizol (Invitrogen). Remaining genomic DNA was eliminated with DNaseI (Roche). 1 µg of total RNA was used for random-priming cDNA synthesis with

SuperScript II (Invitrogen). RT–qPCR was performed using Power SYBR Green in an Applied Biosystems 7900HT Fast qRT–PCR machine. The 2ΔΔCt method was used to quantify amplified fragments. Expression levels were normalized to GAPDH gene.

## Whole-cell extracts, chromatin fractionation, and immunoblots

To prepare whole-cell extracts, cells were trypsinized, collected by centrifugation (290 g/5 min) and resuspended in Laemmli sample buffer (50 mM Tris–HCl pH 6.8, 10% glycerol, 3% SDS, 0.006% w/v bromophenol blue and 5% 2-mercaptoethanol). Cells were sonicated (2 × 30-s in a Branson Digital Sonifier set at 20% amplitude). Chromatin isolation was performed as described (Méndez and Stillman, 2000). Standard methods for SDS-polyacrylamide gel electrophoresis and immunoblot were followed.

## Flow cytometry analysis of DNA content and BrdU incorporation

Cells were pulse-labeled for 30 min with 10 µM BrdU (Sigma) before harvesting, fixed in 70% ethanol, incubated with 2 M HCl (20 min), washed, and incubated with FITC-conjugated anti-BrdU antibody (1 h/37 °C). To monitor DNA content, cells were incubated o/n with 50 µg/ml propidium iodide (PI; Sigma) and 10 µg/ml RNase A (Qiagen). Flow cytometry analyses were performed in a FACS Canto II cytometer (BD, San Jose, CA) and analyzed with FlowJo 9.4 (Tree Star, Ashland, OR). For cell cycle sorting, cells were stained with 5 µg/ml Hoestch 33342 (Invitrogen) for 30 min at 37 °C, resuspended in DMEM + 0.1% FBS and sorted in a BD Influx device (BD, San Jose, CA). For the analysis of cell viability, cells were incubated with 40 nM tetramethyl rhodamine ethylester (TMRE, Sigma) for 10 min at 37 °C, washed in PBS, stained with 1 µg/ml DAPI and analyzed in a FACS Canto II cytometer (BD). Data analysis was performed with FlowJo 9.4 (Tree Star).

## Single-molecule analysis of DNA replication

For origin firing and FR analyses, cells were pulse-labeled with 50 µM CldU (20 min) followed by 250 µM IdU (20 min). Stretched DNA fibers were prepared and analyzed as described (Muñoz et al, 2017). For analysis of origin re-firing, cells were pulse-labeled with 50 µM CldU (2 h) followed by 250 µM IdU (30 min) as described (Neelsen et al, 2013) and DNA fibers were incubated in stringency buffer (10 mM Tris–HCl pH 7.4; 0.4 M NaCl; 0.2% Tween-20; 0.2% NP-40) for 10 min between primary and secondary antibodies (Dorn et al, 2009). Images were obtained in a DM6000 B Leica microscope (HCX PL APO 40×, 0.75 NA objective). DNA fiber thickness was assessed with anti-ssDNA staining to discard overlaps between two different molecules. Single fibers were scored manually.

## Immunofluorescence (IF) analyses

Cells were cultured in polylysine-treated coverslips. To visualize chromatin-bound proteins, soluble factors were pre-extracted with 0.5% Triton X-100 in CSK buffer (10 mM PIPES pH 7.0, 0.1 M NaCl, 0.3 M sucrose, 3 mM MgCl$_2$, 0.5 mM PMSF) for 10 min at 4 °C prior to fixation in 4% PFA (15 min/RT). Coverslips were incubated in blocking solution (3% BSA in PBS + 0.05% Tween 20) for 30 min. Primary (1:100 dilution) and secondary antibody (1:200 dilution) incubations were performed for 1 h/RT. Nuclei were stained with 1 µg/ml DAPI (Sigma) for 1 min. ProLong Gold antifade mounting media was used. Images were acquired in a DM6000 B Leica microscope (HCX PL APO 40×, 0.75 NA objective) and a Leica-TCS SP5-MP confocal microscope (HCX PL APO 40× 1.4 NA oil-immersion objective) and LAS AF v.2.5.1 software. Definiens Developer XD software v.2.5 was used for identification of individual cells and quantification of PCNA, γH2AX or RAD51 intensities and RPA foci. In RS1-treated cells and time-course assays, the intensity of γH2AX, RPA, and RAD51 signals was measured with Cell ProfilerTM software (Broad Institute, https://cellprofiler.org/).

For BrdU native IF, cells were labeled with 10 µM BrdU for 24 h, released in fresh medium and harvested 24 h later. After pre-extraction with CSK buffer (5 min/4 °C) cells were treated with 0.5% Triton X-100 in CSK-stripping buffer (10 mM Tris–HCl pH 7.5, 10 mM NaCl, 3 mM MgCl$_2$, 1% Tween-20, 0.5% sodium deoxicholate) for 6 min at 4 °C. After fixation with 3% PFA (20 min/ 4 °C), cells were incubated with 0.5% Triton X-100 in PBS (5 min/4 °C). Samples were incubated in blocking solution (30 min/ RT). Primary antibody (1:50) was added o/n at 4 °C. Secondary antibody (1:200) was added for 1 h at RT and nuclei were counterstained with DAPI. Images were captured in a DM6000 B Leica microscope (HCX PL APO 40×, 0.75 NA objective).

## Analysis of ChIP-seq data

Rad51 and H2A.Z ChIP-seq data for K562 cells were downloaded from ENCODE (ENCSR524BUE and ENCSR000APC, respectively). ORC2 ChIP-seq data was from Miotto et al (2016); GEO database GSE70165. SNS-seq data for K562 cells was taken from Picard et al (2014); GEO database GSE4618. Heatmaps were generated with computeMatrix (deepTools) using the reference-point utility. Bigwig signal was aligned around peak centers plus/ minus 2.5 kb (10 kb for SNS-seq data) and data was visualized with plotHeatmap (deepTools). Sets of control regions randomized along the genome were prepared with bedtools ShuffleBed. Coincident peaks were obtained with bedtools using the intersect utility. Hg19 assembly of the human reference genome was used.

## Electron microscopy

Cells were collected and resuspended in PBS. Cross-linking was performed with 4,5′,8-trimethylpsoralen (10 µg/ml), followed by irradiation with UV 365 nm monochromatic light (UV Stratalinker 1800; Agilent Technologies). DNA was extracted by cell lysis (1.28 M sucrose, 40 mM Tris–HCl pH 7.5, 20 mM MgCl$_2$, and 4% Triton X-100) and digested (800 mM guanidine–HCl, 30 mM Tris–HCl pH 8.0, 30 mM EDTA pH 8.0, 5% Tween-20, 0.5% Triton X-100) with 1 mg/ml proteinase K (50 °C, 2 h). DNA was purified using chloroform/isoamylalcohol (24:1), precipitated in 0.7 vol of isopropanol, washed with 70% EtOH and resuspended in 200 µL Tris-EDTA buffer. 10 U of PvuII-HF (New England Biolabs) were used to digest 12 µg of DNA for 4–5 h. The digested DNA was purified using the Silica Bead DNA Gel Extraction Kit. For EM analysis, 50–150 ng of DNA were mixed with benzyldimethylalk-ylammonium chloride and formamide, spread on a water surface

and loaded on carbon-coated 400-mesh copper grids as described (Zellweger and Lopes, 2018). DNA was coated with platinum using the high vacuum evaporator MED 020 (BalTec). Imaging was automated using a Talos 120 transmission electron microscope (FEI; LaB6 filament, high tension ≤120 kV) with a bottom mounted CMOS camera BM-Ceta (4k × 4k pixel) and MAPS software (Thermo Fisher Scientific). Molecules of interest were annotated using MAPS Viewer (v.3.16) and images for annotated regions were overlapped and stitched with ForkStitcher (https://github.com/jluethi/ForkStitcher, v. 0.1.1).

### Neutral comet assay

Cells were trypsinized and resuspended in PBS at $10^6$ cells/mL. 20 μL of cell resuspension were mixed with 600 μL of 0.8% w/v Low Melting Point agarose (Lonza) in PBS at 37 °C. 60 μL of cell-LMP mixture were spread onto a comet slide (Trevigen). Slides were incubated for 20 min at 4 °C and submerged o/n in lysis buffer (CometAssay Lysis Solution, Trevigen). Slides were incubated for 1 h at 4 °C in electrophoresis buffer (300 mM sodium acetate, 100 mM Tris pH 8.3) and subjected to electrophoresis in a comet chamber (30 min/21 V/300 mA). Slides were then fixed in 70% ethanol (20 min/4 °C) and dried at 37 °C. DNA was stained for 30 min in the dark using 1:30,000 SYBR Gold (Thermo Fisher Scientific) in 10 mM Tris–HCl pH 7.5, 1 mM EDTA. Microscopy was performed on a Leica DM6 B upright digital research microscope equipped with a DFC360 FX camera at 10× magnification. Images were analyzed using the Open Comet plugin (http://www.cometbio.org) for Fiji.

### Statistical methods

Statistical analyses were performed in Prism v4.0 (GraphPad Software). The number of replicates, cells scored in IFs and specific statistical test applied are indicated in the figure legends. Data acquisition and analysis were performed automatically with specialized software, unless otherwise specified, to ensure unbiased results. No blinding was done. Three or more biological replicas were performed for most assays (2 in the case of Comet and EM analysis). In IF assays, microscopy fields were randomly chosen and >300 cells (or stretched DNA fiber structures) per condition and replicate were scored. Variations among replicates are expected to have normal distributions and equal variances. Data meet the assumptions of the selected test. When two data groups were compared, two-tailed unpaired Student's t-test was used. For comparisons between multiple groups, one-way Anova test and Bonferroni's post-test were applied. Paired-test was used when indicated to reduce sample variability. When the distribution of values in a population is not Gaussian (e.g., track length in stretched DNA fibers), non-parametric Mann–Whitney rank-sum test was used. For multiple comparisons, Kruskal–Wallis test followed by Dunn's post-test was applied.

## Data availability

This study includes no data deposited in external repositories. New biological materials generated in this study are available upon reasonable request.

## Peer review information

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

## Acknowledgements

We thank all members of the DNA Replication and Chromosome Dynamics Groups (CNIO) for discussions, and Patrick Sung (UT Health, San Antonio, TX, USA) for the gift of RAD51 KA mutant. We thank the CNIO Flow Cytometry and Confocal Microscopy core Units (Biotechnology Programme) for excellent technical support. Research in JM lab was supported by grants BFU2016-80401-R, PID2019-106707-RB, and PID2022-142177-NB funded by MCIN/AEI/10.13039/501100011033 and by "ERDF A way of making Europe". Work in ML lab was supported by SNSF Project grant 310030_189206. DG-A is supported by EMBO post-doctoral Fellowship 1135-2021. SM was the recipient of a CNIO Friends post-doctoral contract (2020 call).

## Author contributions

**Sergio Muñoz:** Conceptualization; Investigation; Methodology; Writing—original draft; Writing—review and editing. **Elena Blanco-Romero:** Investigation. **Daniel González Acosta:** Investigation. **Sara Rodriguez-Acebes:** Investigation. **Diego Megías:** Formal analysis; Methodology. **Massimo Lopes:** Funding acquisition; Investigation. **Juan Méndez:** Conceptualization; Supervision; Funding acquisition; Writing—original draft; Writing—review and editing.

## Disclosure and competing interests statement

The authors declare no competing interests.

# Expanded View Figures

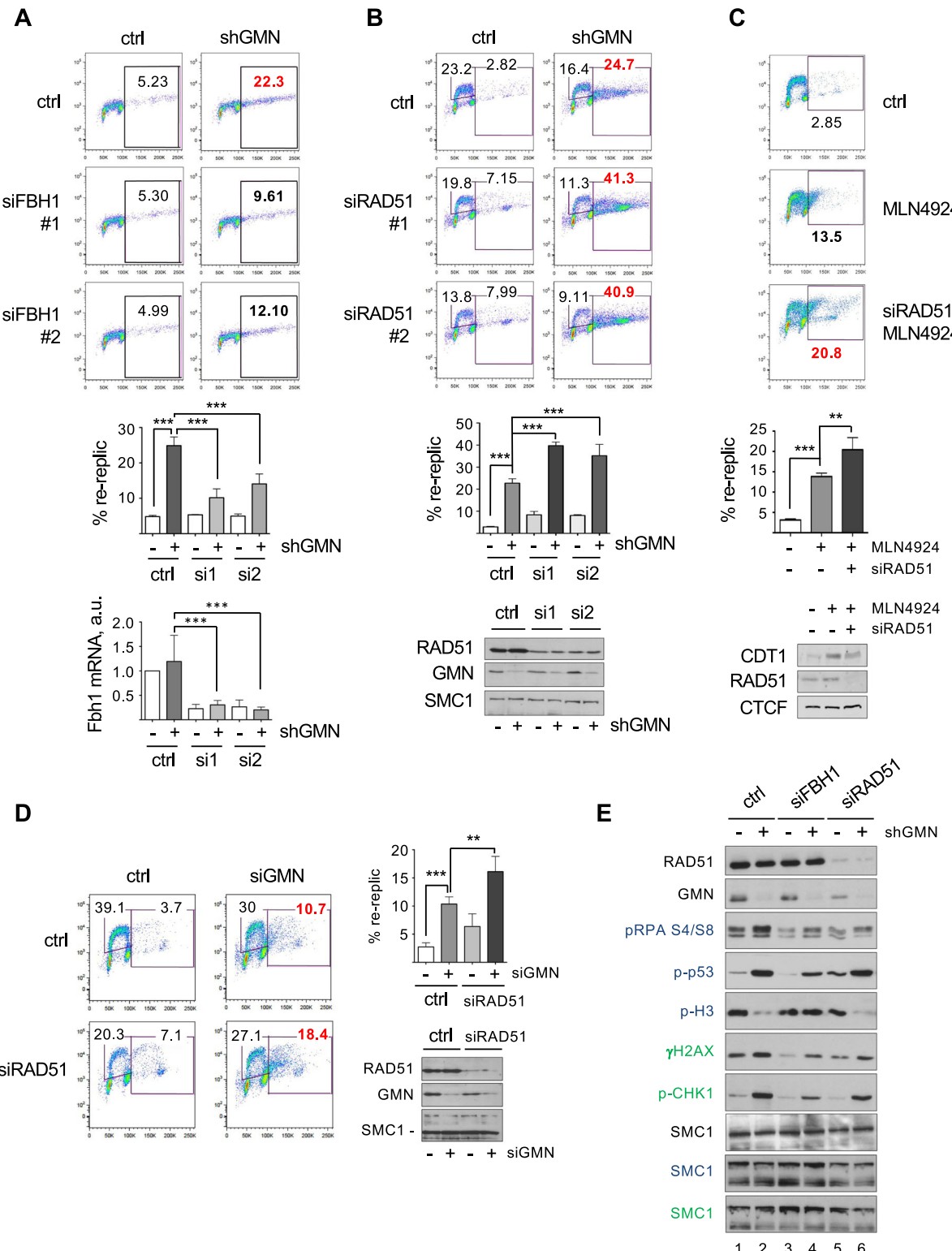

**Figure EV1. Opposing effects of RAD51 and FBH1 on DNA re-replication (related to Fig. 1).**

(A) Top, analysis of re-replication in HCT116-shGMN cells treated with or without shGMN and transfected with the indicated FBH1 siRNAs for 72 h. Gates indicate cells with over-replicated DNA. Middle histogram shows the percentage (mean and SD) of cells undergoing re-replication. $n = 3$ replicates. ***$p < 0.001$ (one-way Anova and Bonferroni's post-test). Bottom histogram shows fold-change (mean and SD) of Fbh1 mRNA levels relative to control. $n = 3$ replicates. ***$p < 0.001$; one-way Anova and Bonferroni's post-test. (B) Same as (A), in HCT116-shGMN cells transfected with the indicated RAD51 siRNAs for 72 h. S-phase cells are included in the small gate, and cells with over-replicated DNA are included in the large gate. Middle histogram shows the percentage (mean and SD) of cells undergoing re-replication. $n = 3$ replicates. ***$p < 0.001$; one-way Anova and Bonferroni's post-test. Immunoblots show the levels of RAD51 and GMN proteins. SMC1 is shown as loading control. (C) Analysis of re-replication in HCT116-shGMN cells in the absence or presence of 250 nM MLN4924 and transfected with RAD51 siRNA for 72 h when indicated. Middle histogram shows the percentage (mean and SD) of cells undergoing re-replication. $n = 3$ replicates. ***$p < 0.001$; **$p < 0.01$ (one-way Anova and Bonferroni's post-test). Immunoblots show the levels of RAD51 and CDT1 proteins. CTCF is shown as loading control. (D) Analysis of re-replication in U2OS cells transfected with siGMN and/or siRAD51 for 72 h as indicated. Top right, histogram shows the percentage (mean and SD) of cells undergoing re-replication. $n = 3$ replicates. ***$p < 0.001$; **$p < 0.01$ (one-way Anova and Bonferroni's post-test). Bottom right, immunoblot detection of RAD51 and GMN proteins. SMC1 is included as loading control. (E) Immunoblot detection of the indicated proteins in HCT116-shGMN cells from the experiment shown in main Fig. 1D. SMC1 levels are shown as loading control. The three separate gels used to analyze this experiment are indicated by the legend color (black/green/blue). Source data are available online for this figure.

**Figure EV2. HR factors and MUS81 do not regulate DNA re-replication (related to Fig. 2).**

(A) Immunoblot detection of GMN in cells from the experiment described in main Fig. 2A, B. TUBULIN is shown as loading control. (B) Analysis of re-replication in HCT116-shGMN cells treated with or without shGMN for the indicated time. Histogram shows the percentage (mean and SD) of cells undergoing re-replication. $n = 3$ replicates. ***$p < 0.001$; n.s., not significant (one-way Anova and Bonferroni's post-test). The levels of GMN are shown by immunoblot. Tubulin is shown as loading control. (C) Percentage of cells displaying positive staining for γH2AX and chromatin-bound RAD51 in samples from main Fig. 2A. $n = 3$ replicates. **$p < 0.01$ (two-tailed, unpaired Student's t test). (D) Same as (C) for chromatin-bound RPA and RAD51 signals. $n = 3$ replicates. ***$p < 0.001$ (two-tailed, unpaired Student's t test). (E) Immunoblot detection of the RAD51AP1 and p-RPA32 (S4/S8) in HCT116-shGMN cells transfected with siRNA51AP1 and treated with 10 μM camptothecin (CPT) for 24 h as indicated. CTCF is shown as loading control. The two separate gels used to analyze this experiment are indicated by the legend color (black/blue). (F) Immunoblot levels of p21 protein in HCT116-shGMN cells treated with 10 μM B02 and/or 0.1 mM doxorubicin (DXR) for 24 h as indicated. SMC1 is shown as loading control. (G) Percentage of cells in each phase of the cell cycle, corresponding to the experiment shown in main Fig. 2E. $n = 3$ replicates. **$p < 0.01$ (two-tailed, unpaired Student's t test). Source data are available online for this figure.

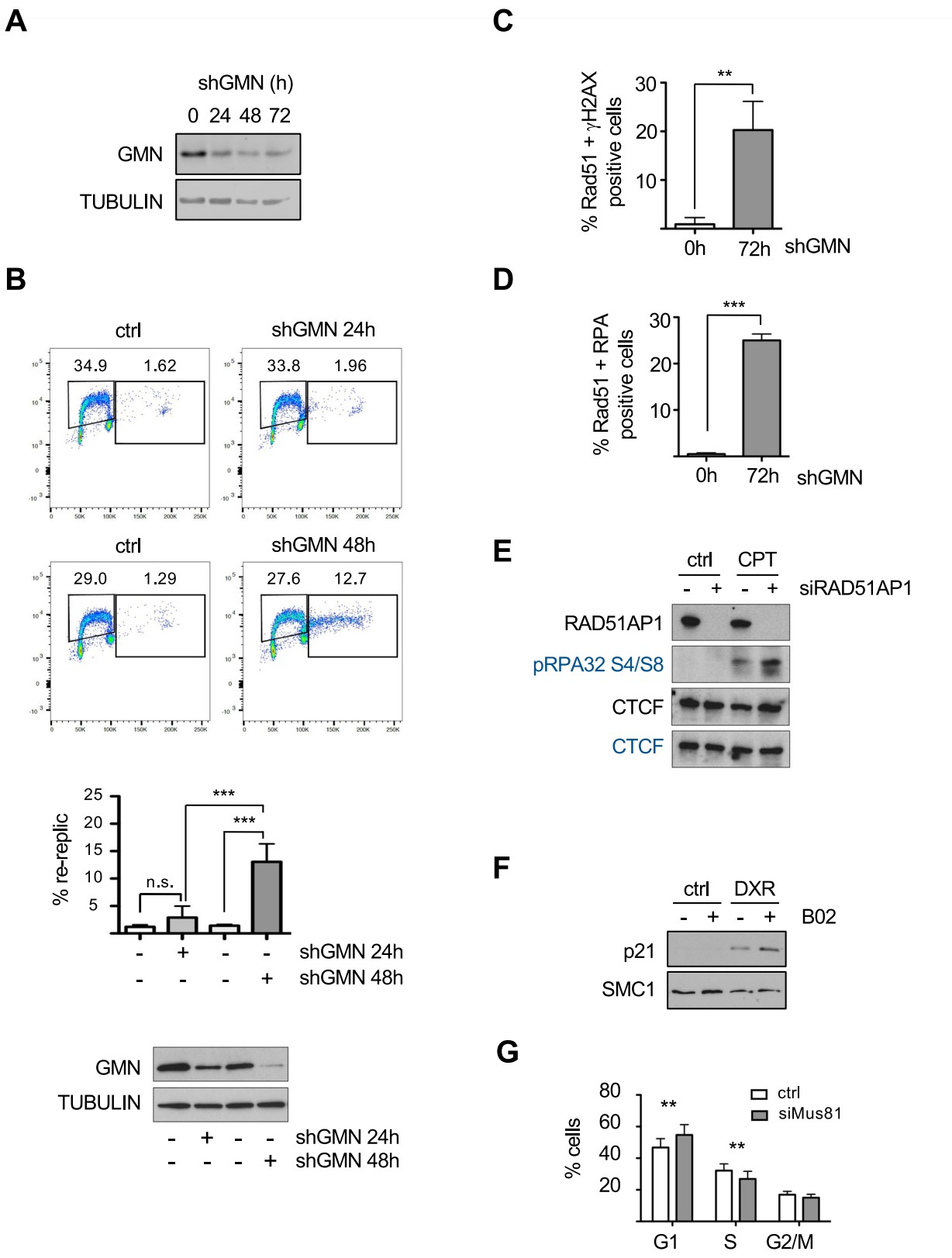

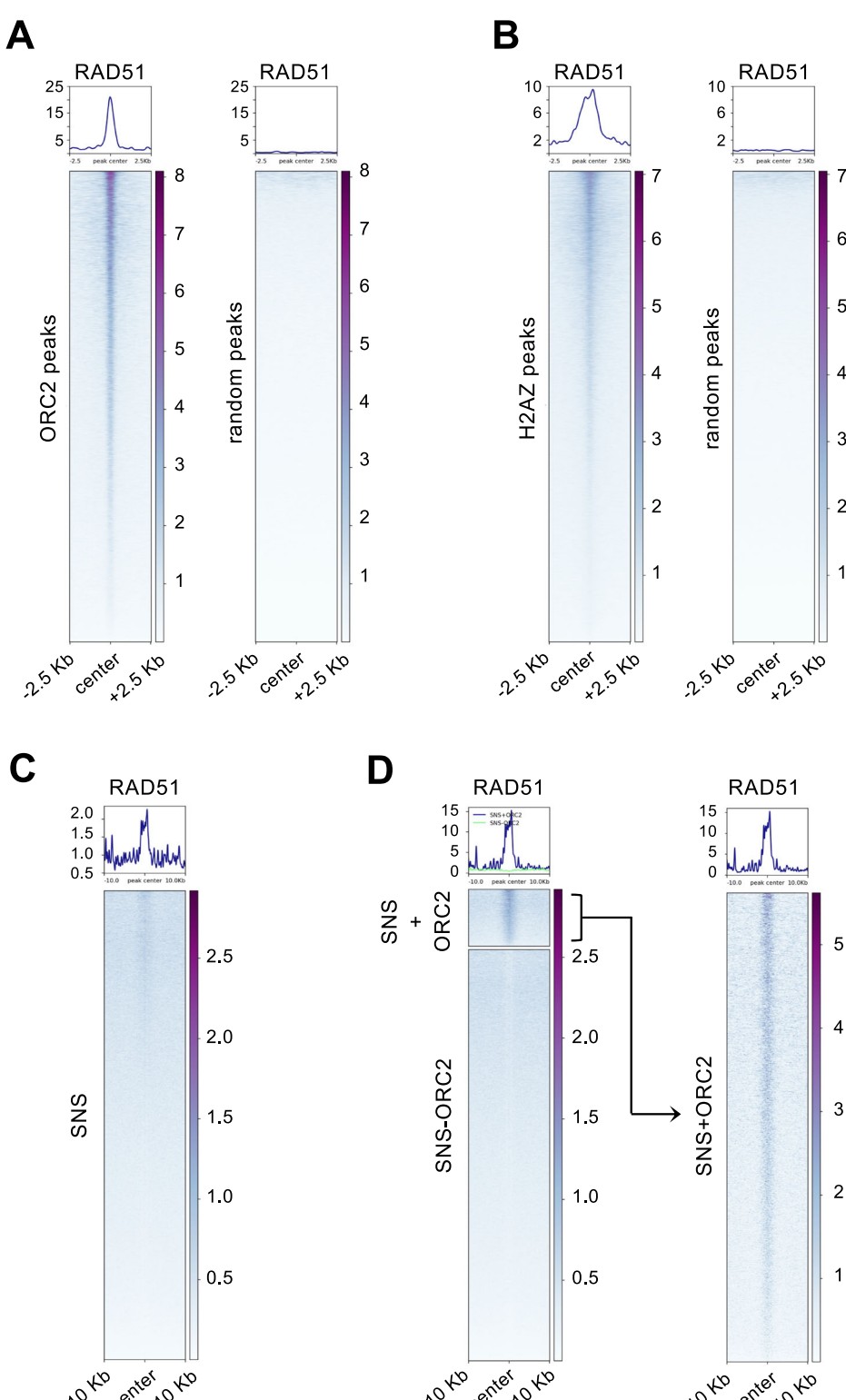

**Figure EV3. RAD51 enrichment around replication origins (related to Fig. 3).**

(A) Heatmap distribution of RAD51 ChIP-seq reads around ORC2-binding sites (left) or an equal number of randomized genomic positions (right). (B) Same as (A), showing the distribution of RAD51 around H2A.Z binding sites (left) or randomized genomic positions (right). (C) Heatmap distribution of RAD51 around all SNS-seq peaks. (D) Left, heatmap distribution of RAD51 around SNS-seq peaks stratified according to the simultaneous detection of ORC2 (SNS + ORC2 and SNS-ORC2 respectively). Right, blow-up of the distribution of RAD51 around SNS-seq peaks coincident with ORC2-binding sites.

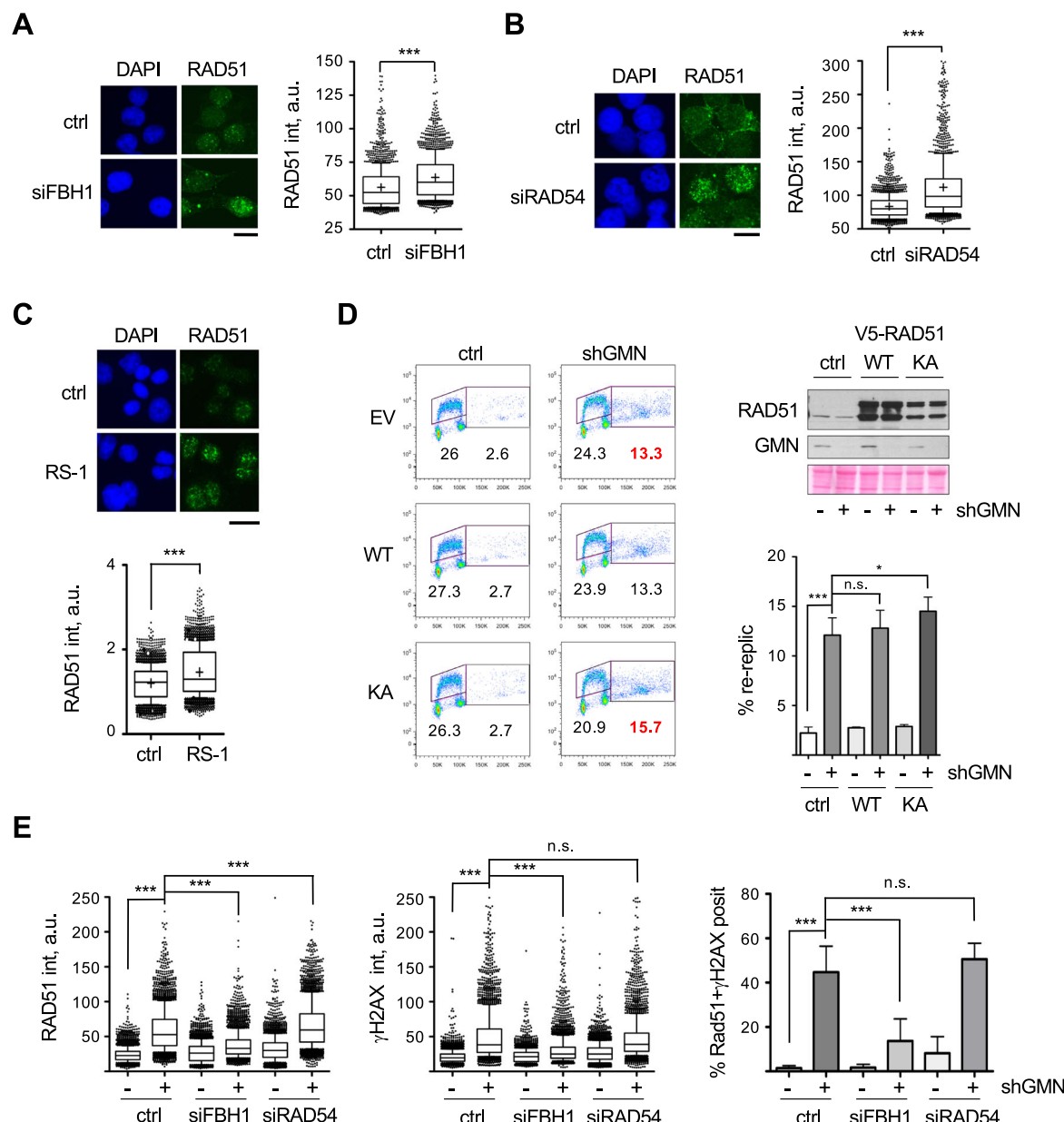

**Figure EV4. Chromatin-bound RAD51 restricts DNA re-replication (related to Fig. 3).**

(A) Representative IF images of HCT116-shGMN cells transfected when indicated with siFBH1 siRNA for 72 h, pre-extracted and immunostained with a different RAD51 antibody than the one used in main Fig. 3C. DNA was counterstained with DAPI (blue). Bar, 15 μm. Right, box and whiskers plot showing the distribution of chromatin-bound RAD51 intensity. Data from two experiments were pooled. $n = 1000$ cells per condition and replica. ***$p < 0.001$ (Mann–Whitney test). (B) Same as (A), in HCT116-shGMN cells transfected when indicated with RAD54 siRNA for 72 h. Bar, 15 μm. Data from two different experiments were pooled. $n = 1000$ cells per condition and replica. ***$p < 0.001$ (Mann–Whitney test). (C) Same as (A) and (B), in HCT116-shGMN cells treated with 7.5 μM RS1 for 24 h. Bar, 15 μm. Data from 4 different experiments were pooled. $n = 800$ cells per condition and replica. ***$p < 0.001$ (Mann–Whitney test). (D) Analysis of re-replication in HCT116-shGMN cells treated with or without shGMN and transfected with the indicated V5-tagged RAD51 variant for 72 h. Ctrl, empty vector; WT, V5-RAD51-WT; KA, V5-RAD51-K133A. Top right, immunoblot detection of RAD51 and GMN proteins. Red Ponceau-S staining of the membrane is shown as loading control. Bottom right, percentage (mean and SD) of cells undergoing re-replication. $n = 3$ replicates. ***$p < 0.001$; *$p < 0.05$; n.s., not significant (repeated measures Anova and Bonferroni's post-test). (E) Box and whiskers plots showing the distribution of RAD51 (left) and γH2AX (center) intensity in pre-extracted HCT116-shGMN cells treated with or without shGMN and transfected with the indicated siRNAs for 72 h. ctrl, siFBH1 and siRAD54 (-shGMN) samples are the same as in main Fig. 3C. Data from 3 different experiments are pooled. $n = 1000$ cells per condition and replica. ***$p < 0.0001$ (Kruskal–Wallis test and Dunn's post-test). Right histogram shows the percentage of cells displaying positive staining for γH2AX and chromatin-bound RAD51 in the indicated samples. $n = 3$. ***$p < 0.001$ (one-way Anova and Bonferroni's post-test). Box plots in (A–C,E): Boxes are drawn from the 25th to the 75th percentile. The central horizontal line indicates the median value. Whiskers are drawn from the 10th to the 90th percentile and the rest of values are drawn as individual dots (mimima, 0th percentile, maxima 100th percentile). In (A–C), the mean values are drawn as "+". Source data are available online for this figure.

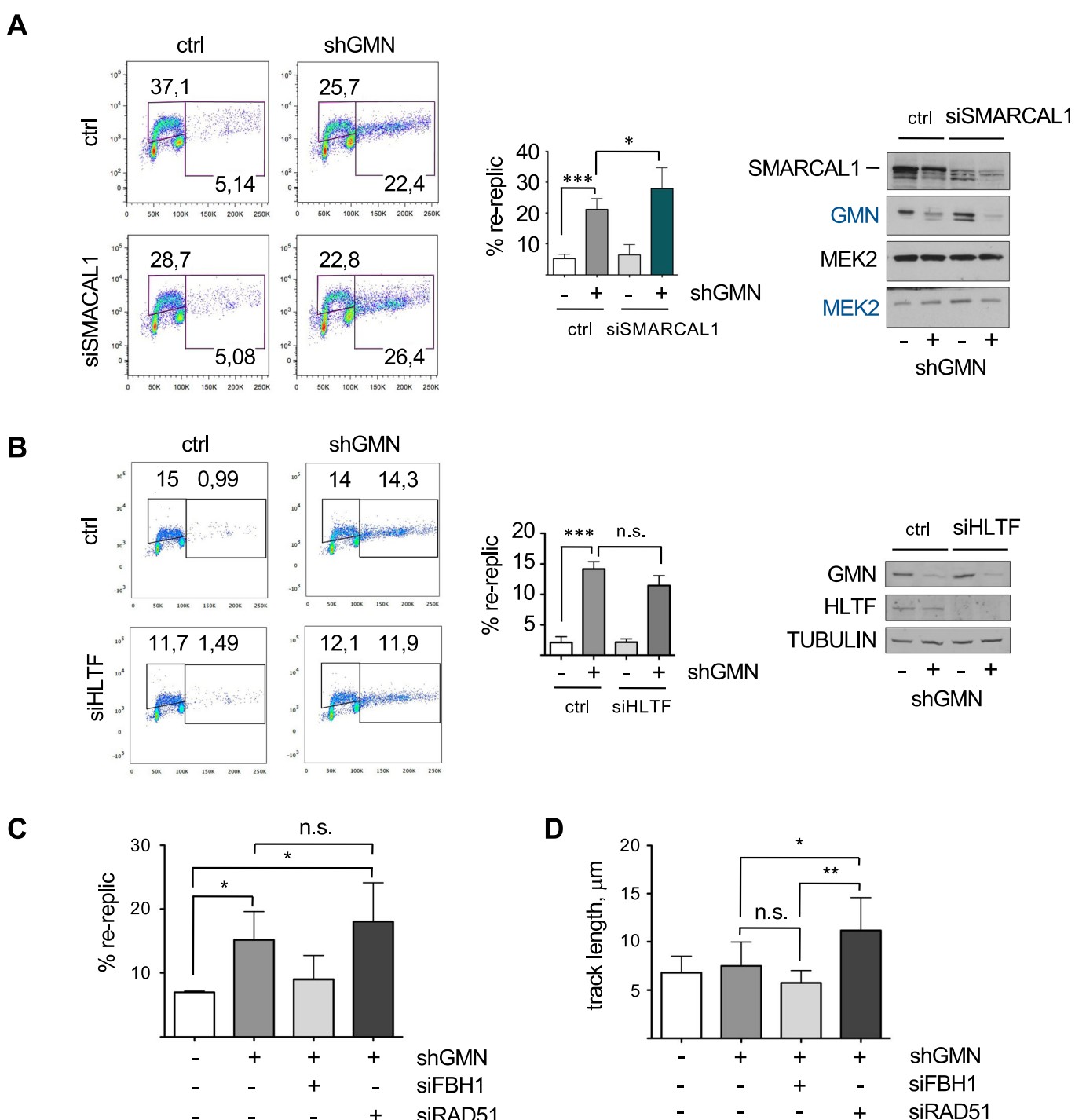

**Figure EV5.   RAD51 blocks the progression of re-replication forks (related to Fig. 4).**

(A) Analysis of re-replication in HCT116-shGMN cells grown with or without shGMN and transfected when indicated with siSMARCAL1 for 72 h. Histogram shows the percentage (mean and SD) of cells undergoing re-replication. $n = 3$ replicates. ***$p < 0.001$; *$p < 0.05$ (repeated measures Anova and Bonferroni's post-test). Right, immunoblot detection of SMARCAL1 and GMN protein levels. MEK2 is shown as loading control. The two separate gels used to analyze this experiment are indicated by the legend color (black/blue). (B) Same as (A), in HCT116-shGMN cells transfected with siHLTF when indicated. $n = 3$ replicates. ***$p < 0.001$; n.s. not significant (repeated measures Anova and Bonferroni's post-test). Immunoblot shows HLTF and GMN protein levels, with TUBULIN as loading control. (C) Percentage (mean and SD) of re-replicated tracks relative to the total number of replicative structures from the experiment shown in main Fig. 4G. $n = 3$ replicates, with >394 replicative structures scored per condition and replicate. *$p < 0.05$; n.s., not significant (one-way Anova and Bonferroni's post-test). (D) Median length (mean and SD) of re-replicated tracks in samples from the experiment shown in main Fig. 4G. $n = 3$ replicates. **$p < 0.01$; *$p < 0.05$; n.s., not significant (one-way Anova and Bonferroni's post-test). Source data are available online for this figure.

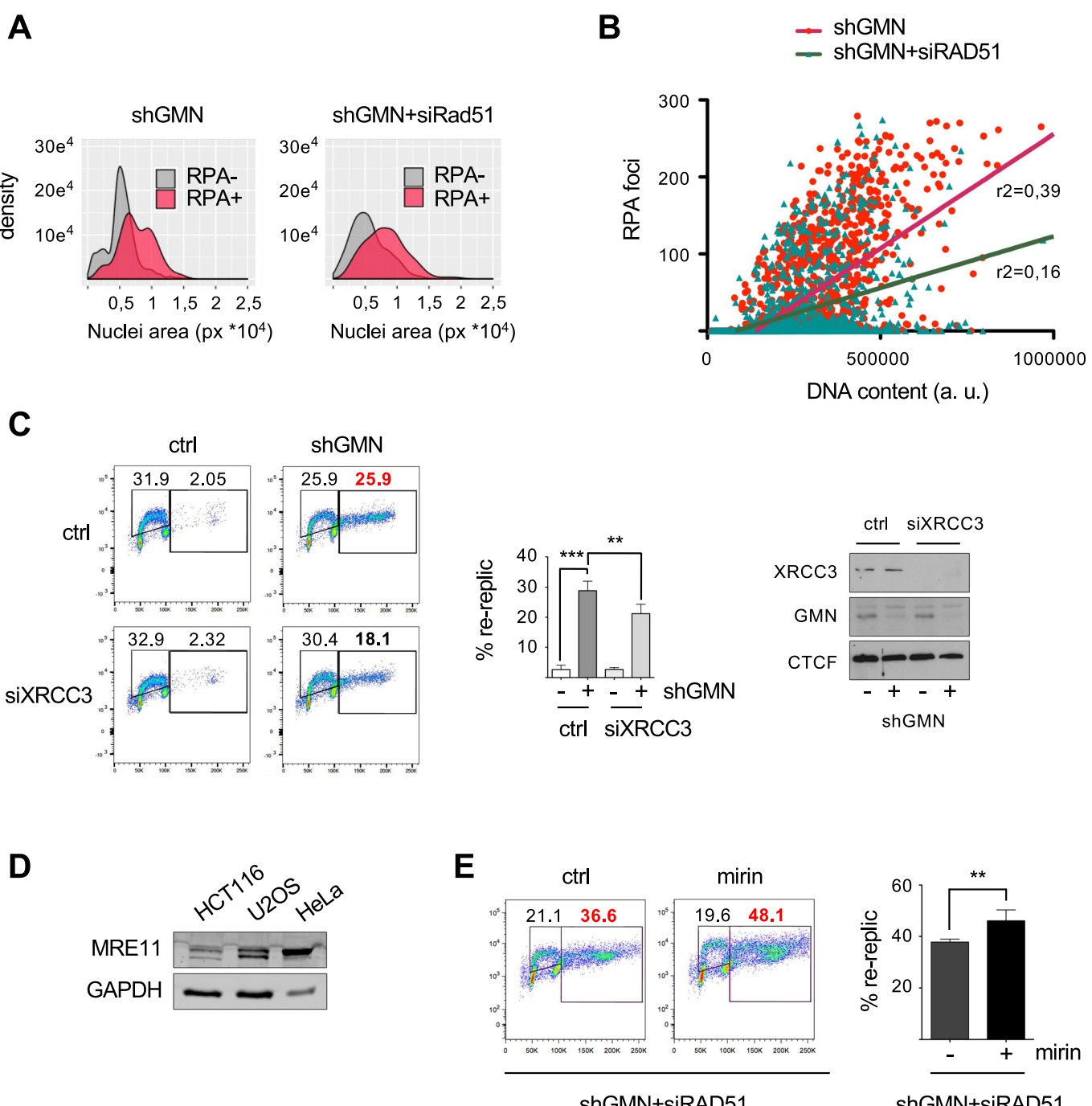

**Figure EV6.** Discontinuous DNA synthesis during re-replication (related to Fig. 5).

(**A**) Distribution of nuclear area in RPA-negative and RPA-positive cells (with or without siRAD51) derived from the experiment shown in main Fig. 5B. Cells with >10 RPA foci were considered positive. 450 cells from each condition were analyzed. One representative experiment (out of four) is shown. (**B**) Correlation between number of RPA foci and DNA content in the cells used in main Fig. 5B ($p < 0.001$ in Spearman non-parametric correlation test in both conditions). >1450 cells from each condition were analyzed. One representative experiment (out of four) is shown. The difference between slopes was statistically significant ($p < 0.001$ in linear regression test); R2 coefficients are indicated. (**C**) Analysis of re-replication in HCT116-shGMN cells grown with or without shGMN and transfected when indicated with XRCC3 siRNA for 72 h. Histogram shows the percentage (mean and SD) of cells undergoing re-replication. $n = 3$ replicates. ***$p < 0.001$; **$p < 0.01$ (one-way Anova and Bonferroni's post-test). Right, immunoblot detection of XRCC3 and GMN protein levels, with CTCF as loading control. (**D**) Immunoblot detection of MRE11 protein in different cell lines, with GAPDH as reference. (**E**) Analysis of re-replication in HCT116-shGMN cells transfected with siRAD51 (72 h). When indicated, 10 μM mirin was added for 24 h. Histogram shows the percentage (mean and SD) of cells undergoing re-replication. $n = 4$ assays. **$p < 0.01$ (Student's t-test). Source data are available online for this figure.

