## [Peer Review File · The EMBO Journal]

RAD51 restricts DNA over-replication from re-activated origins

Juan Mendez, Sergio Muñoz, Elena Blanco-Romero, Daniel Gonzalez-Acosta, Sara Rodriguez-Acebes, Diego Megías, and Massimo Lopes

Corresponding author(s): Juan Mendez (jmendez@cniio.es)

Review Timeline:

Submission Date:	14th Mar 23
Editorial Decision:	16th Apr 23
Revision Received:	11th Dec 23
Editorial Decision:	8th Jan 24
Revision Received:	11th Jan 24
Accepted:	12th Jan 24

Editor: Hartmut Vodermaier

Transaction Report:

Dr. Juan Mendez
CNIO
Molecular Oncology Programme
Melchor Fernandez Almagro, 3
Madrid E-28029
Spain

16th Apr 2023

Re: EMBOJ-2023-114022
RAD51 restricts DNA over-replication from re-activated origins

Dear Juan,

Thank you again for submitting your study on re-replication restriction after initiation to The EMBO Journal. I have now received the reports of three expert referees, copied below for your information. As you will see, all referees consider your findings interesting and potentially important, but at the same time raise a number of substantive concerns that indicate that not all parts of the study are already sufficiently conclusive. Should you be able to adequately address these criticisms, we would be happy to consider a revised version of the study further for publication.

Since it is our policy to consider only a single round of major revision and therefore important to fully answer to all comments at the time of resubmission, I would invite you to consider the reports together with your coworkers, and to prepare a tentative response letter detailing how each of the raised criticisms/queries might be answered/clarified. On the basis of this response, we could then discuss the requirements for a successful revision already during the early stages of the revision, e.g. via email or a follow-up video call. I should add that we could also offer extension of the default three-months revision period if needed, with our 'scoping protection' (meaning that competing work appearing elsewhere in the meantime will not affect our considerations of your study) remaining of course valid also throughout this extension.

Detailed information on preparing, formatting and uploading a revised manuscript can be found below and in our Guide to Authors. Thank you again for the opportunity to consider this work for The EMBO Journal, and I look forward to hearing from you in due time.

With kind regards,

Hartmut

9) Digital image enhancement is acceptable practice, as long as it accurately represents the original data and conforms to community standards. If a figure has been subjected to significant electronic manipulation, this must be clearly noted in the figure legend and/or the 'Materials and Methods' section. The editors reserve the right to request original versions of figures and the original images that were used to assemble the figure. Finally, we generally encourage uploading of numerical as well as gel/blot image source data; for details see: embopress.org/page/journal/14602075/authorguide#sourcedata

At EMBO Press, we ask authors to provide source data for the main manuscript figures. Our source data coordinator will contact you to discuss which figure panels we would need source data for and will also provide you with helpful tips on how to upload and organize the files.

In the interest of ensuring the conceptual advance provided by the work, we recommend submitting a revision within 3 months (15th Jul 2023). Please discuss the revision progress ahead of this time with the editor if you require more time to complete the revisions. Use the link below to submit your revision:

Link Not Available

Referee #1:

In this manuscript, in combination with geminin-depletion, Munoz et al knocked down various proteins involved in DNA transduction, and screened factors that affected the extend of re-replication. Authors show that depletion of Rad51 enhances the levels of re-replication, while knock-down of its obstructors, Fbh1 and Rad54 have an opposite effect. In addition, they present that PrimPol mediates a DNA synthesis on template formed from re-initiated origins, leading to the formation of ssDNA gaps, which recruits nuclease MRE11 to degrade re-replicated DNA. In Rad51 depleted cells, frequency of fork reversal is decreased and replication fork rate is increased, but origin-refiring rate is not. Based on these results, they propose a novel role of Rad51, performed together with PrimPol and Mre11, in preventing re-replication by restricting the fork progression and eliminating re-replicated DNA. The finding presented here contrasts to the past works mostly addressing blocking re-replication at origin re-licensing, and represents a novel control to prevent re-replication. I found this is an interesting story and have following comments and concerns which should be addressed before acceptance by the EMBO Journal.

major concerns:

1. Authors examined the role of Rad51 in the regulation of re-replication in geminin-depleted cells. Re-replication can be also induced in various situations, such as when Emi1 or Cdt2 is depleted, or Cdt1 over-expressed. The re-replication brought about in geminin-depleted cells appears to be different from the other cases, since re-replication in this case begins from G2 phase (Klotz-Noack et al. 2012). To mention as title says "RAD51 restricts DNA over-replication from re-activated origins", authors need to show that Rad51 has a similar role in other cases.

2. The immunostaining of Rad51 in Fig. 3 and EV2 shows pictures of asynchronously growing cells. Readers will be interested in the chromatin association of Rad51 during re-replication. To demonstrate how Rad51 engages in the process, authors need to show the time course analysis of Rad51 staining after geminin depletion. In addition, it is important to see if Rad51 is co-localized with stress markers such as gamma H2AX and RPA and to see how Rad51 staining will be affected when Fbh1 or Mre11 is depleted.

3. If Rad51 restricts re-replication, it is expected that high-expression of Rad51 would reduce re-replication and thus increase cell survival. However, Figure EV2F shows expression of WT Rad51 has no effect on the extent of re-replication. How do authors explain this? In addition, to show the importance of Rad51 pathway in restricting re-replication, it is required to see and compare the cell survival rates in various conditions which affect Rad51 function and levels of re-replication, such as Fbh1 depletion, Rad51 over-expression or depletion etc.

4. Authors propose that Mre11 associates with the ssDNA gaps produced by PrimPol-primed DNA synthesis and then it eliminates re-replicated DNA and prevents expansion of re-replication (positive effect of PrimPol on restricting re-replication). On the other hand, silencing of PrimPol reduces the extent of re-replication (Fig.5D), suggesting that PrimPol has a negative effect on restricting re-replication. I wonder what happens in the absence of PrimPol? Authors need to describe a mechanism explaining how re-replication is reduced in the absence of PrimPol.

minor concerns:

5. It looks like that Rad51 depletion alone induces substantial amount of re-replication and DNA breaks (Fig. 1D and 1F). Is Rad51 important for preventing re-replication in a normal cell cycle?

6. In a genetic screen, downregulation of a helicase, probably WRN, greatly enhanced nuclear size (Fig.1A and Table EV1). Since WRN also controls fork reversal, how do authors consider its involvement in restricting re-replication?

7. "1C" and "2C" could be better changed to "2C" and "4C".

Referee #2:

In this study, Muñoz and colleagues characterize a novel role for RAD51 in mitigating the extent of re-replication from inappropriately re-fired origins. The authors performed a targeted genetic screen to identify modifiers of re-replication by monitoring nuclear size after geminin depletion. Top hits included RAD51, a recombinase involved in homologous recombination and fork protection, and a RAD51 antagonist, FBH1. Rad51 depletion enhanced the amount of excess DNA/nucleus whereas FBH1 depletion enhanced Rad51 chromatin binding and restricted excess DNA/nucleus.

Strengths include targeting multiple regulators of Rad51 and related proteins using various siRNA depletions and, where available, pharmacological inhibitors. The authors rigorously distinguish fork extension from additional origin firing by DNA fiber analysis. Taken together, the dataset supports a model in which the presence of Rad51 on chromatin restrains replication fork progression in geminin-depleted cells while having little effect on origin re-firing itself. The use of mutants and manipulating other homologous recombination genes separates Rad51's effect on re-replication fork extension from a role for HR per se. Additional experiments shed some light on some unique aspects of DNA metabolism at re-replicating forks such as dependent on PrimPol and sensitivity to MRE11.

Overall, this study is comprehensive, uses multiple quantitative assays and is well-presented. There are a few questions and opportunities for improvement that we list here:

1. It would be relatively straightforward to repeat one or more central findings in another cell line.
2. Some of the important results are the absence of effects from depleting other genes such as RAD54, MUS81, etc. or inhibiting HR. The authors should comment on how they ensured the treatments had the desired effect. In other words, how do they know the targeted activity is low enough that they would have seen an effect if there were one? As presented, some results hinge solely on the reduced band in the western blot, and some depletions appear more profound than others.
3. The authors cite Alexander et al. as reporting similar Rad51-mediated effects on fork progression at amplifying loci in *Drosophila*. The authors of that study did not make that claim and instead, report no effect of the Rad51 homolog mutation (spnA) on amplifying fork progression - see for example, Figure 4 of that study. The authors should clarify their interpretation of the 2015 paper.
4. What is the ultimate fate of cells with more or less re-replication fork progression? Rad51 is an essential gene even in cells with normal geminin levels. (Figures 1D and EV1B suggest that Rad51 depletion alone can enhance endogenous re-replication which was also not mentioned.)

Minor (At the authors' discretion)

- a) >2C DNA content for the flow cytometry data should be >4C.
- b) The text (methods or legends) should include how individual cells and DNA molecules were selected for counting
- c) Make it clear on the figure/legend that the data in Figure 3B is chromatin because "biochemical fractionation" may not be clear to all readers.
- d) For the model in figure 6, define colors and line thickness of the arrows for clarity

Referee #3:

In this manuscript, Muñoz and colleagues report the characterization of a novel pathway that limits re-replication in human cells. Re-replication occurs when replication origins are activated more than once per cell cycle. A large body of evidence indicates that re-replication is prevented by multiple mechanisms acting at the level of initiation, essentially by preventing the licensing of origins during the S phase. Here, the authors screened for factors that either reduce or increase re-replication in cells lacking Geminin, a key repressor of origin licensing. They identified a novel mechanism that limits re-replication downstream of initiation through a process involving the RAD51 recombinase. They also report that re-replication is enhanced in the absence of FBH1 and RAD54, two RAD51 antagonists; and upon inhibition of MRE11, a nuclease involved in the degradation of nascent DNA at reversed forks and ssDNA gaps. They propose an original mechanism in which RAD51 does not preclude origin re-firing but hinders the progression of re-replicated forks, either by physically blocking forks or by promoting fork reversal. In this model, multiple rounds of re-replicated fork arrest and restart would limit the extent of re-replication and promote the degradation of re-replicated DNA by MRE11. These findings are important because they explain the slow elongation rate during re-replication and define a novel and important mechanism that limits re-replication downstream of the known mechanisms that prevent origin re-licensing. Overall, the data are of high quality and the results are presented in a clear and concise manner. However, the authors need to address several important issues listed below to strengthen their argument.

Major issues:

1. Fig. 3E: The effect of RS-1 is not very convincing. The authors would make their case stronger by showing that it affects RAD51 binding (as in Fig. 3C) at this concentration and/or that it decreases re-replication in a dose-dependent manner.
2. Fig. 4B: The authors conclude that these results "... provide direct evidence that chromatin-bound RAD51 in S-phase limits re-replication by hindering the progression of re-replication forks." Fig. 4E indicates that the % of re-replicated tracks does not exceed 20 %, so why are all forks faster upon RAD51 depletion? In this experiment, 80% of the measured tracks should correspond to normal forks, and should not be affected by RAD51. Could it be that re-replication generates an (ATR-dependent?) signal that slows down all forks in the genome? If so, why are these forks faster than forks from control cells? These are important questions that need to be addressed prior to publication.
3. Fig. 4D: The authors use overlapping IdU and CldU signals as an indication of re-replication events. In the experimental design of the experiment, the IdU track is not located in the middle of the CldU track. Is there a reason for this? Wouldn't one expect the same origin to fire? Also, a white signal could also result from the overlay of two different fibers, which is quite common in DNA fiber spreading experiments. Does this explain the relatively high percentage of re-replication observed in control cells (~8%, Fig. 4E)? This technical issue does not call into question the main conclusions of the experiment, but should be addressed in the text.
4. The proposed model indicates that RAD51 could either act directly by slowing down fork progression or indirectly by promoting fork reversal. Wouldn't it be possible to discriminate between these possibilities? From this respect, it is interesting that the depletion of HLTF and ZRANB3 (but not SMARCAL1) increases nuclear volume (indicative of re-replication) in the screen, but to a lower extent than RAD51. It would be interesting to compare the effect of the depletion of RAD51 and ZRANB3 or HLTF on fork speed during re-replication. This could be a way to determine the relative contribution of the direct and indirect effects of RAD51 on fork speed.
5. Twenty years ago, the Caldecott lab reported that RAD51 and its paralogs slow down forks in the presence of damaged DNA (PMID: 12718895). This reference should be cited, along with PMID: 32669601. In addition, the potential role of RAD51 paralogs in re-replication should be discussed, even though they were not included in the screen. How do re-replication forks compare to normal forks encountering DNA lesions?
6. Along the same line, the authors previously reported that rapid cell cycles in stem cells also lead to replication stress and rely on PrimPol for growth (PMID: 26876348 ; PMID: 36152632). It may be worth discussing similarities between re-replication (Geminin depleted cells) and replication after a short G1 phase (stem cells).
7. It has been reported that MRE11 is mutated in HCT116 cells and other mismatch repair deficient cancer cell lines (PMID: 11850399). Since MRE11 is involved in many of the processes described in this study, the status of MRE11 should be addressed.

Minor issues:

1. Fig. 1C: A low level of re-replication is detectable in Ctrl siRAD51 cells, which is consistent with earlier publications (cited in the introduction) showing that re-replication occurs spontaneously in normal cells, but the authors do not comment on that. Whether RAD51 depletion aggravates the consequences of spontaneous re-replication in normal cells is an interesting observation that would be worth mentioning.
2. Fig. EV1C: Indicate that the anti p-RPA antibody is directed against phospho-S4/S8.
3. Fig. EV2: The presence of RAD51 near ORC2 peaks in ChIP-seq data from the Snyder lab is difficult to understand. Indeed, the experiment was performed in human K562 cells that are not supposed to re-replicate. Moreover, since not all ORC2 sites correspond to active replication origins, the presence of RAD51 near ORC2 binding sites could be due to other reasons. Along the same line, the correlation with initiation sites through H2AZ enrichment is very indirect (Long, 2020). It would have made more sense to compare RAD51 enrichment with initiation sites as determined by SNS-seq, for example from PMID: 32209126 for HCT116 cells or PMID: 34108027 for K562 cells.
4. Page 8 (1st paragraph): Provide more information in text on how the experiments shown in Fig. 3A-C were performed (shGMN cells, cell sorting...).
5. Fig. 4B: The fork speed measured for control cells seems to be fairly low. According to DNA combing analyses (PMID: 30796221), it should be around 1.1 kb/min in HCT116 cells. The difference could come from the standard conversion factor used here ($1 \mu\text{M} = 2.59 \text{ kb}$). Where does this factor come from? Is it applicable here? Otherwise, the authors could express distances in μm , rather than in kb.
6. Fig EV4C: Indicate R^2

POINT-BY-POINT RESPONSE LETTER**Reviewer #1: [AU RESPONSES in blue font]**

In this manuscript, in combination with geminin-depletion, Munoz et al knocked down various proteins involved in DNA transduction, and screened factors that affected the extend of re-replication. (...) The finding presented here contrasts to the past works mostly addressing blocking re-replication at origin re-licensing, and represents a novel control to prevent re-replication. I found this is an interesting story and have following comments and concerns which should be addressed before acceptance by the EMBO Journal.

We thank the reviewer for the thorough assessment of our study and the positive comments. We have addressed the specific comments as follows:

1. Authors examined the role of Rad51 in the regulation of re-replication in geminin-depleted cells. Re-replication can be also induced in various situations, such as when Emi1 or Cdt2 is depleted, or Cdt1 over-expressed. The re-replication brought about in geminin-depleted cells appears to be different from the other cases, since re-replication in this case begins from G2 phase (Klotz-Noack et al. 2012). **To mention as title says "RAD51 restricts DNA over-replication from re-activated origins", authors need to show that Rad51 has a similar role in other cases.**

We would like to point out that the analysis of re-replicated tracks using stretched DNA fibers indicates that at least some re-replication events caused by loss of GMN take place in S phase. In any case, we agree about the relevance of confirming the role of RAD51 when DNA re-replication is induced in a different manner.

To this aim we have used MLN4924, a potent inhibitor of the NEDD8-activating enzyme that results in CDT1 protein stabilization and significant amounts of re-replication in S-phase (Fu et al, 2021). 9h of MLN4924 treatment in HCT116 cells caused detectable re-replication in up to 13.8% of the cells. When MLN4924 was combined with Rad51 downregulation, this percentage went up to 20.4% (average of three replicates), confirming the restrictive role of RAD51 in re-replication. This experiment is included in the revised manuscript as **Fig EV1C** and mentioned in Results (p. 5-6).

2. The immunostaining of Rad51 in Fig. 3 and EV2 shows pictures of asynchronously growing cells. Readers will be interested in the chromatin association of Rad51 during re-replication. To demonstrate how Rad51 engages in the process, authors need to **show the time course analysis of Rad51 staining after geminin depletion**. In addition, it is important to **see if Rad51 is co-localized with stress markers such as gamma H2AX and RPA** and to see **how Rad51 staining will be affected when Fbh1 or Mre11 is depleted**.

As suggested, we have performed time-course experiments to monitor the binding of RAD51 to chromatin following GMN downregulation and have also checked RS markers. The interpretation of these analysis, however, is slightly complicated by the dual roles of RAD51: protection of nascent DNA and HR-mediated DNA repair. As expected, the amount of RAD51 on chromatin increased as re-replication (and DNA damage) accumulated. In fact, by 72 h post-shGMN >20% of the cells were double-positive for γ H2AX and chromatin-bound RAD51 (or chromatin-bound RPA and RAD51). We interpret that many RAD51 molecules are recruited to damaged DNA to mediate HR repair of DSBs. These results are included in the revised manuscript (new **Figure 2A-B** and **Figure EV2A-D**) and discussed in p. 6 (Results).

We have also tested how FBH1 depletion affected RAD51 staining and DNA damage markers. In the new analyses we have included RAD54, because both FBH1 and RAD54 restrict the binding of RAD51 to undamaged DNA but affect HR-directed DNA repair in opposite ways: loss of FBH1 may result in hyperrecombination (Simandlova et al, 2013) while loss of RAD54 causes defective HR repair (Heyer et al, 2006). Consistent with these antecedents, we found that FBH1 downregulation reduced both DNA re-replication and DNA damage, while RAD54 downregulation limited re-replication without reducing DNA damage, indicating that DNA breaks could not be repaired by HR. These results, shown in new **Fig EV4E** and mentioned in the revision (Results, p. 8-9 and Discussion, page 16), contribute to separate the roles of RAD51 in the control of re-replication and HR-mediated DNA repair.

3. If Rad51 restricts re-replication, it is expected that high-expression of Rad51 would reduce re-replication and thus increase cell survival. However, Figure EV2F shows **expression of WT Rad51 has no effect on the extent of re-replication. How do authors explain this?**

This is an interesting point; however, whether transient overexpression of WT RAD51 should impact re-replication is debatable. Re-replication could be affected only marginally if the extra RAD51 protein is not loaded onto DNA after the first round of replication. Ectopically expressed RAD51 should overcome the regulatory function of RAD54 and FBH1, in a sufficient number of cells to facilitate its detection by flow cytometry. On the other hand, RAD51 K133A mutant may form mixed filaments with WT RAD51 and achieve a dominant-negative effect, as reported for DNA damage repair (Kim et al, 2012). Therefore, a limited amount of mutant protein might render a stronger effect than transient overexpression of WT RAD51 as in the experiment shown in **Fig EV4D**. This point could be further addressed in a future study through the generation of stable cell lines that overexpress RAD51 (WT or mutant versions) with different abilities to bind DNA, but this falls beyond the scope of our current study. Given the limitations with WT RAD51 overexpression outlined above, we considered that testing the effects of FBH1 or RAD54 depletion, which affect the stability and release of RAD51 from the DNA, were more informative approaches.

In addition, to show the importance of Rad51 pathway in restricting re-replication, it is required to see and **compare the cell survival rates in various conditions which affect Rad51 function and levels of re-replication**, such as Fbh1 depletion, Rad51 over-expression or depletion etc.

In response to this point, we have now monitored the activation of apoptosis and levels of cell death upon GMN loss in control, siFBH1-treated and siRAD51-treated cells. Interventions that reduced re-replication favored cell viability. Conversely, downregulation of RAD51 that increased over-replicated DNA and prevented DNA damage repair, resulted in viability loss. These new data are included in the revised manuscript as **Figure 1G-H** (Results, p. 6).

4. Authors propose that Mre11 associates with the ssDNA gaps produced by PrimPol-primed DNA synthesis and then it eliminates re-replicated DNA and prevents expansion of re-replication (positive effect of PrimPol on restricting re-replication). On the other hand, silencing of PrimPol reduces the extent of re-replication (Fig.5D), suggesting that PrimPol has a negative effect on restricting re-replication. I wonder what happens in the absence of PrimPol? Authors **need to describe a mechanism explaining how re-replication is reduced in the absence of PrimPol**.

We understand that this issue may have caused confusion, and we have addressed with additional experiments and clarifications in the text. In our model, the key role of PRIMPOL is to promote continuation of DNA synthesis at re-replication forks that were stalled and reversed in the presence of RAD51. The function of PRIMPOL as an antagonist of fork reversal has been well established in other contexts (e.g. Quinet et al, 2020; Jacobs et al, 2022). In our study, PRIMPOL inhibition limits the progression of re-replication forks, reducing the global levels of re-replication. We have now combined the downregulation of PRIMPOL and RAD51. The rationale is that in the absence of RAD51, fork stalling would be minimal and PRIMPOL should be less relevant. Our new results strongly confirm this idea: siPRIMPOL prevented re-replication only when RAD51 was present (**new Fig 5D** in the revised manuscript, discussed in p. 11).

Additional experiments carried out to address a comment from **reviewer 3** also shed light into this point. We have tested the effect of XRCC3, a RAD51 paralog that also participates in the restart of stalled forks (Berti et al, 2020). Downregulation of XRCC3 also restricted the extent of re-replication, underscoring the notion that re-replication fork progression requires events of fork restart. This is discussed in p. 11 and new **Figure EV6C** in the revised manuscript.

In summary, the main contribution of PRIMPOL in this context is to restart stalled re-replication forks, favoring their extension. A side effect of this role, however, is the generation of ssDNA gaps that serve as entry points for MRE11 nuclease that may later contribute to degrade the excess of DNA.

Other (minor) concerns:

5. It looks like that Rad51 depletion alone induces substantial amount of re-replication and DNA breaks (Fig. 1D and 1F). **Is Rad51 important for preventing re-replication in a normal cell cycle?**

All reviewers have mentioned this point, which is now mentioned and discussed in the manuscript (page 6, Results and page 13, Discussion). Our interpretation is that origin re-firing occurs sporadically in normally cycling cells (evidence to this effect is mentioned in the Introduction). Given their higher levels of initiator proteins, origin re-firing may be more frequent in cancer cell lines, making the role of RAD51 even more relevant.

6. In a genetic screen, downregulation of a helicase, probably WRN, greatly enhanced nuclear size (Fig.1A and Table EV1). Since WRN also controls fork reversal, how do authors consider its involvement in restricting re-replication?

We noticed this effect and indeed tested the effect of WRN downregulation. However, siWRN induces a major block in G2 that is responsible for the increment in nuclear size, without discernable signs of DNA re-replication. Given this side effect, we have not further pursued WRN as a candidate hit.

7. "1C" and "2C" could be better changed to "2C" and "4C".

This has been changed according to the reviewer's suggestion.

Reviewer #2

In this study, Muñoz and colleagues characterize a novel role for RAD51 in mitigating the extent of re-replication from inappropriately re-fired origins. (...) Overall, this study is comprehensive, uses multiple quantitative assays and is well-presented.

We thank the reviewer for the positive assessment of our study and the specific comments and questions.

There are a few questions and opportunities for improvement that we list here:

1. It would be relatively straightforward **to repeat one or more central findings in another cell line.**

We have now confirmed the key observation that siRAD51 stimulates re-replication using U2OS cells in which GMN was downregulated with siRNA. This is shown as **Figure EV1D** and described in p. 6 in the revised manuscript.

2. Some of the important results are the absence of effects from depleting other genes such as RAD54, MUS81, etc. or inhibiting HR. The authors should **comment on how they ensured the treatments had the desired effect.** In other words, how do they know the targeted activity is low enough that they would have seen an effect if there were one?

We agree with the reviewer about the need to include control assays to make sure that the indicated treatments had a cellular effect, particularly when the extent of re-replication was not affected.

Downregulation of FBH1 or RAD54 significantly reduced DNA re-replication levels and increased the amount of chromatin-bound RAD51. For those treatments that did not affect DNA re-replication, the following controls are now included:

- (i) siRAD51AP1: levels of DNA damage (p-RPA S4/S8) were higher following RAD51AP1 downregulation and exposure to camptothecin, as described (Parplys et al, 2015). This is shown in new **Fig EV2E** and mentioned in p. 7.
- (ii) B02: the abundance of p21 protein was increased in the presence of B02 and doxorubicin, as described (Schürmann et al, 2021). This is shown in new **Fig EV2F** and mentioned in p. 7.
- (iii) siMUS81: the cell cycle pattern was altered, leading to an accumulation of cells in G1, as described (Naim et al, 2013). This is shown in new **Fig EV2G** and mentioned in the text (p. 7).

3. The authors cite Alexander et al. as reporting similar Rad51-mediated effects on fork progression at amplifying loci in *Drosophila*. The authors of that study did not make that claim and instead, report no effect of the Rad51 homolog mutation (*spnA*) on amplifying fork progression - see for example, Figure 4 of that study. The authors should **clarify their interpretation of the 2015 paper**.

We did not think that we were misrepresenting the previous study, as the sentence that we wrote in the original manuscript (“*the progression of re-replication forks in D. melanogaster follicle cells is enhanced in the absence of RAD51 homologs*”) was taken almost verbatim from the abstract of the referred work: “*Conversely, **we show that fork progression is enhanced in the absence of both Drosophila Rad51 homologs, spindle-A and spindle-B, revealing homologous recombination is active...***” However, as the reviewer points out, their interpretation was different from ours.

We have rewritten this section to mention both the original interpretation and our complementary (not exclusive) suggestion. Alexander et al (2015) argued that RAD51 homologs slow re-replication because HR repair of DSBs generated by re-replication is slower than microhomology-mediated end-joining repair. In the light of the results shown in our manuscript, we simply suggest that, in addition to the effect reported in the original article, the RAD51 homologs (*spindle-A* and *-B*) may also directly block the progression of re-replication forks in *Drosophila* follicle cells. This is discussed in the revised article, p. 14.

4. What is the ultimate fate of cells with more or less re-replication fork progression? Rad51 is an essential gene even in cells with normal geminin levels. (Figures 1D and EV1B suggest that Rad51 depletion alone can enhance endogenous re-replication which was also not mentioned.)

Both points were shared by other reviewers. As mentioned in the response to reviewer 1, we have tested the levels of non-viable (apoptotic and dead) cells following the induction of re-replication and this is shown in new **Figure 1G-H** and mentioned in p. 6 in the revision.

We have also discussed the possible role of Rad51 in preventing re-replication in a normal cell cycle (page 6, Results and page 13, Discussion).

Minor (at the authors' discretion)

a) >2C DNA content for the flow cytometry data should be >4C.

Changed as suggested.

b) The text (methods or legends) should include how individual cells and DNA molecules were selected for counting.

Methods (and, when needed, Figure legends) have been modified to reflect this.

c) Make it clear on the figure/legend that the data in Figure 3B is chromatin because "biochemical fractionation" may not be clear to all readers.

This has been clarified in the revised text (p. 7) and in **Figure 3B** (panel and legend).

d) For the model in figure 6, define colors and line thickness of the arrows for clarity

Original Figure 6 marked leading and lagging strands with different colors, and also separated first and second rounds of replication. However, we now see that too many colors complicate the message and in the revision we have adopted a simpler color scheme that is explained in the Figure legend. Line thickness has also been unified. We believe that with these changes, the model in **Figure 6** has gained in clarity.

Reviewer #3:

In this manuscript, Muñoz and colleagues report the characterization of a novel pathway that limits re-replication in human cells (...) They propose an original mechanism in which RAD51 does not preclude origin re-firing but hinders the progression of re-replicated forks, either by physically blocking forks or by promoting fork reversal (...) These findings are important because they explain the slow elongation rate during re-replication and define a novel and important mechanism that limits re-replication downstream of the known mechanisms that prevent origin re-licensing. Overall, the data are of high quality and the results are presented in a clear and concise manner. However, the authors need to address several important issues listed below to strengthen their argument.

We thank the reviewer for the positive assessment of our study and the specific suggestions for improvement.

Major issues:

1. Fig. 3E: The effect of RS-1 is not very convincing. The authors would **make their case stronger by showing that it affects RAD51 binding** (as in Fig. 3C) at this concentration and/or that it decreases re-replication in a dose-dependent manner.

We agree that the treatment with RS-1 does not yield a very large effect, but we included it in the manuscript as supporting data for other, more conclusive experiments. We have included this clarification in the text (page 8) and linked it to **Figure EV4C** that showed the effect of RS-1 on RAD51 binding.

2. Fig. 4B: The authors conclude that these results "... provide direct evidence that chromatin-bound RAD51 in S-phase limits re-replication by hindering the progression of re-replication forks." Fig. 4E indicates that the % of re-replicated tracks does not exceed 20 %, so why are all forks faster upon RAD51 depletion? In this experiment, 80% of the measured tracks should correspond to normal forks, and should not be affected by RAD51. **Could it be that re-replication generates an (ATR-dependent?) signal that slows down all forks in the genome?** If so, why are these forks faster

than forks from control cells? These are important questions that need to be addressed prior to publication.

This is an interesting conceptual point. While part of the difference between changes in fork rate and percentage of re-replicated tracks could be attributed to technical reasons, the activation of a systemic fork slowdown in response to DNA damage and replication stress (induced by origin re-firing) is a very plausible scenario. According to published literature, RAD51 is necessary to slow down forks in the presence of DNA damage (Henry-Mowatt et al, 2003) and its depletion promotes fork progression in stressed conditions (Zellweger et al, 2015; Berti et al, 2020). These effects are linked to the ability of RAD51 to promote fork reversal. Reducing fork reversal with olaparib in cells treated with DNA damaging agents increased fork rate (Ray Chaudhuri et al, 2012, Berti et al 2013; Zellweger et al, 2015). In fact, DNA damage induces a global response mediated by ATR and RAD51 to stop and reverse active forks, even those that have not encountered damaged DNA directly (Mutreja et al, 2018).

In the light of these antecedents, it is conceivable that re-replication-induced DNA damage causes a systemic response that slows down “regular” forks” and promotes reversal. In agreement with this notion, the percentage of reversed structures in shGMN-treated cells is approximately 2-fold higher than the percentage of re-replicated tracks in the same conditions (**Fig 4D** and **EV5C**).

In a new experiment incorporated to the revision, inhibition of global fork reversal with olaparib restored the fork rate of re-replicating cells almost to control levels (new **Fig 4E**). Therefore, inhibition of fork reversal is responsible, at least in part, for the higher fork rate observed in RAD51 and GMN co-depleted cells. However, because the effect of RAD51 downregulation on fork rate was higher than that of olaparib, we conclude that RAD51 has an additional role over re-replication, which we attribute to a direct block to re-replication forks.

While it is likely that ATR mediates the slowdown and reversal response, testing its participation in the context of origin re-firing poses a fundamental problem. Because ATR inhibitors (ATRi) trigger new origin firing, to evaluate their effect specifically on

forks requires the use of CDC7 inhibitors (Mutreja et al, 2018). But in our cellular system, CDC7 inhibitors would automatically prevent origin re-firing and the formation of re-replication forks.

3. Fig. 4D: The authors use overlapping IdU and CldU signals as an indication of re-replication events. In the experimental design of the experiment, the IdU track is not located in the middle of the CldU track. Is there a reason for this? Wouldn't one expect the same origin to fire?

Analysis of re-activated origins in stretched DNA fibers is a technically demanding variation with limited capacity for fine-tuning. However, this observation from the reviewer is accurate and could have several explanations. First, it is possible that we have captured origins displaying asymmetric fork progression (i.e. one of the two forks is progressing faster than the other) which is very common in situations of stress. This fork asymmetry would make the signals corresponding to origin reactivation look off-center. Another plausible possibility (not exclusive with the previous one) is that the second initiation event takes place from another origin located in the same "initiation zone" than the first one. In this case, the track would not necessarily be located in the center of the original replicated area.

Also, a white signal could also result from the overlay of two different fibers, which is quite common in DNA fiber spreading experiments. Does this explain the relatively high percentage of re-replication observed in control cells (~8%, Fig. 4E)? This technical issue does not call into question the main conclusions of the experiment, but should be addressed in the text.

While there is a technical risk of counting tracks derived from overlapping fibers, we pay special attention to minimize this effect. An anti-ssDNA staining is used to identify and discard overlapping fibers, based on signal thickness. Also, fiber tracks are counted manually instead of relying on automatic software. Besides, the levels of re-replication in unchallenged cancer cells are similar to those described in an independent report (Dorn et al, 2009). These technical aspects are now mentioned in the revised Materials and Methods section (p. 19).

4. The proposed model indicates that RAD51 could either act directly by slowing down fork progression or indirectly by promoting fork reversal. Wouldn't it be possible to discriminate between these possibilities? From this respect, is interesting that the depletion of HLTF and ZRANB3 (but not SMARCAL1) increases nuclear volume (indicative of re-replication) in the screen, but to a lower extent than RAD51. It would be interesting **to compare the effect of the depletion of RAD51 and ZRANB3 or HLTF on fork speed during re-replication**. This could be a way to determine the relative contribution of the direct and indirect effects of RAD51 on fork speed.

Our model is that both direct hindering of re-replication fork progression and induction of fork reversal contribute to restrict the extension of re-replication. The contribution of fork reversal to the overall change in fork rate has been addressed with olaparib, as discussed above (point 2). Besides, we have found that olaparib treatment increased the percentage of cells undergoing re-replication (new **Figure 4F**; p. 10).

Olaparib counteracts fork reversal through hyperactivation of RecQL1 (Berti et al 2013), independently of the translocases involved. In this regard, depletion of SMARCAL1, but not HLTF, rendered similar effects (new **Figure EV5A-B**; p. 10; the contribution of ZRANB3 could not be tested because we could not validate the downregulation of the protein).

We found that global inhibition of fork reversal induced less over-replication than RAD51 depletion. These observations support an additional role of RAD51 in hindering the progression of re-replication forks. The complexity of this dual control exerted by RAD51 is presented in the revised schematic model in **Figure 6**.

5. Twenty years ago, the Caldecott lab reported that RAD51 and its paralogs slow down forks in the presence of damaged DNA (PMID: 12718895). This reference should be cited, along with PMID: 32669601. In addition, **the potential role of RAD51 paralogs in re-replication should be discussed**, even though they were not included in the screen. How do re-replication forks compare to normal forks encountering DNA lesions?

To address this question, we have tested the effect of Rad51 paralog XRCC3. As showed in the study cited by the reviewer (Berti et al, 2020), XRCC3 mediates restart of DNA synthesis at reversed forks without participating in the process of fork reversal itself. Therefore, its downregulation could parallel the loss of PRIMPOL, also involved in DNA synthesis restart after fork stalling. Indeed, following XRCC3 downregulation, the average percentage of cells undergoing re-replication upon GMN loss was reduced. This is now shown in **Figure EV6C** and mentioned in p. 11.

While it is difficult to compare directly how re-replication and regular forks react to lesions/obstacles, the available data obtained by downregulation of PRIMPOL, XRCC3, or olaparib do not suggest any fundamental differences in terms of requirements for reversal and restart. Mechanisms that promote fork restart favor the accumulation of re-replication, while mechanisms that prevent it, also restrict re-replication.

6. Along the same line, the authors previously reported that rapid cell cycles in stem cells also lead to replication stress and rely on PrimPol for growth (PMID: 26876348 ; PMID: 36152632). It **may be worth discussing similarities between re-replication (Geminin depleted cells) and replication after a short G1 phase (stem cells)**.

Indeed, both re-replicating cells and stem cells experience high levels of RS and DNA damage, and the reactivation of their stalled forks largely relies on PRIMPOL. Whereas in stem cells PRIMPOL reactivates regular forks, upon origin re-firing it mainly reactivates re-replicated forks. This parallelism is now mentioned in the Discussion (p.15).

7. It has been reported that MRE11 is mutated in HCT116 cells and other mismatch repair deficient cancer cell lines (PMID: 11850399). Since MRE11 is involved in many of the processes described in this study, **the status of MRE11 should be addressed**.

We thank the referee for pointing out this fact. The mutation alluded (Giannini et al, 2002) affects MRE11 gene splicing and the global levels of the protein. Still, about half of the transcripts yield a normal, functional protein. We have now confirmed that while MRE11 in HCT116 cells is a bit less abundant than in U2OS or HeLa cells, the protein is readily detected and its levels are affected by mirin (**Fig 5E** and **EV6D**; p. 12 of revised text).

Minor issues:

1. Fig. 1C: (...) Whether RAD51 depletion aggravates the consequences of spontaneous re-replication in normal cells is an interesting observation that would be worth mentioning.

This was also suggested by the other two referees. The role of RAD51 during re-replication in an unchallenged S phase is now mentioned in the Results and Discussion sections (p. 6 and p. 13).

2. Fig. EV1C: Indicate that the anti p-RPA antibody is directed against phospho-S4/S8.

This is now indicated as suggested (this experiment is now **Fig EV1E**).

3. Fig. EV2: The presence of RAD51 near ORC2 peaks in ChIP-seq data from the Snyder lab is difficult to understand. (...) It would have made more sense to compare RAD51 enrichment with initiation sites as determined by SNS-seq, for example from PMID: 32209126 for HCT116 cells or PMID: 34108027 for K562 cells.

Indeed it is interesting to compare RAD51 ChIP-seq with SNS-seq data. We have used the published SNS-seq data for K562 cells (Picard et al, 2014) to avoid comparing genome-wide data from two different cell lines. A modest enrichment of RAD51 positions was found around SNS peaks when the complete dataset was used. The enrichment of RAD51 was much more clear around the subset of SNS-seq peaks that coincide with ORC2 signals (presumably corresponding to highly active origins). These correlations support the idea that at least a fraction of total RAD51 is located at or around origins of replication. These new analyses are shown in **Fig EV3C-D** and discussed in page 8 of the revised manuscript.

Nevertheless, we would like to emphasize that we do not argue that RAD51 binds to origins in order to prevent re-firing. Instead, we suggest that its binding to newly synthesized DNA after origin firing (to protect nascent DNA from nucleases, as described) contributes to prevent re-replication extension in the event of origin re-activation.

4. Page 8 (1st paragraph): Provide more information in text on how the experiments shown in Fig. 3A-C were performed (shGMN cells, cell sorting...).

Text has been amended to address this comment (p. 7-8).

5. Fig. 4B: The fork speed measured for control cells seems to be fairly low. According to DNA combing analyses (PMID: 30796221), it should be around 1.1 kb/min in HCT116 cells. The difference could come from the standard conversion factor used here ($1 \text{ \AA}\mu\text{M} = 2.59 \text{ kb}$). Where does this factor come from? Is it applicable here? Otherwise, the authors could express distances in μm , rather than in kb.

In our extensive experience with the stretched DNA fiber technique, we are aware that technical factors like lab humidity and temperature may influence the degree of stretching and the absolute numbers of fork progression rate. These effects likely contribute to the different values obtained in different reports. Besides, stretched DNA fibers and classic DNA combing involve different attachments of DNA molecules to the slide surface.

In any case, we agree that using an universal conversion factor derived from a 1998 report is probably no longer needed. We have changed the figures, figure legends, and Methods section to plot distances in μm rather than Kb.

6. Fig EV4C: Indicate R^2

Coefficients of determination (R^2) have been added, as suggested.

LITERATURE CITED IN THE POINT-BY-POINT RESPONSE LETTER

- Alexander JL, Beagan K, Orr-Weaver TL, McVey M. 2016. Multiple mechanisms contribute to double-strand break repair at rereplication forks in *Drosophila* follicle cells. *Proc Natl Acad Sci U S A*. 113:13809-13814.
- Berti M, Ray Chaudhuri A, Thangavel S, Gomathinayagam S, Kenig S, Vujanovic M, Odreman F, Glatter T, Graziano S, Mendoza-Maldonado R, et al. 2013. Human RECQ1 promotes restart of replication forks reversed by DNA topoisomerase I inhibition. *Nat Struct Mol Biol*. 20:347-54.
- Berti M, Teloni F, Mijic S, Ursich S, Fuchs J, Palumbieri MD, Krietsch J, Schmid JA, Garcin EB, Gon S, et al. 2020. Sequential role of RAD51 paralog complexes in replication fork remodeling and restart. *Nat Commun*. 11:3531.
- Dorn ES, Chastain PD, Hall JR, Cook JG. 2009. Analysis of re-replication from deregulated origin licensing by DNA fiber spreading. *Nucleic Acid Res* 37: 60-69.
- Fu H, Redon CE, Thakur BL, Utani K, Sebastian R, Jang SM, Gross JM, Mosavarpour S, Marks AB, Zhuang SZ, et al. 2021. Dynamics of replication origin over-activation. *Nat Commun*. 12:3448.
- Giannini G, Ristori E, Cerignoli F, Rinaldi C, Zani M, Viel A, Ottini L, Crescenzi M, Martinotti S,

- Bignami M, et al. 2002. Human MRE11 is inactivated in mismatch repair-deficient cancers. *EMBO Rep.* 3:248-54.
- Henry-Mowatt J, Jackson D, Masson JY, Johnson PA, Clements PM, Benson FE, Thompson LH, Takeda S, West SC, Caldecott KW. 2003. XRCC3 and Rad51 modulate replication fork progression on damaged vertebrate chromosomes. *Mol Cell.* 11:1109-17.
- Heyer WD, Li X, Rolfsmeier M, Zhang XP. 2006. Rad54: the Swiss Army knife of homologous recombination? *Nucleic Acids Res.* 34:4115-25.
- Jacobs K, Doerdelmann C, Krietsch J, González-Acosta D, Mathis N, Kushinsky S, Guarino E, Gómez-Escolar C, Martinez D, Schmid JA, et al. 2022. Stress-triggered hematopoietic stem cell proliferation relies on PrimPol-mediated repriming. *Mol Cell.* 82:4176-88.e8.
- Kim TM, Ko JH, Hu L, Kim SA, Bishop AJ, Vijg J, Montagna C, Hasty P. 2012. RAD51 mutants cause replication defects and chromosomal instability. *Mol Cell Biol.* 32:3663-80.
- Mason JM, Dusad K, Wright WD, Grubb J, Budke B, Heyer WD, Connell PP, Weichselbaum RR, Bishop DK. 2015. RAD54 family translocases counter genotoxic effects of RAD51 in human tumor cells. *Nucleic Acids Res.* 43:3180-96.
- Mutreja K, Krietsch J, Hess J, Ursich S, Berti M, Roessler FK, Zellweger R, Patra M, Gasser G, Lopes M. 2018. ATR-Mediated Global Fork Slowing and Reversal Assist Fork Traverse and Prevent Chromosomal Breakage at DNA Interstrand Cross-Links. *Cell Rep.* 24:2629-42.e5.
- Naim V, Wilhelm T, Debatisse M, Rosselli F. 2013. ERCC1 and MUS81-EME1 promote sister chromatid separation by processing late replication intermediates at common fragile sites during mitosis. *Nat Cell Biol.* 15:1008-15.
- Parpys AC, Zhao W, Sharma N, Groesser T, Liang F, Maranon DG, Leung SG, Grundt K, Dray E, Idate R, et al. 2015. NUCKS1 is a novel RAD51AP1 paralog important for homologous recombination and genome stability. *Nucleic Acids Res.* 43:9817-34.
- Picard F, Cadoret JC, Audit B, Arneodo A, Alberti A, Battail C, Duret L, Prioleau MN. 2014. The spatiotemporal program of DNA replication is associated with specific combinations of chromatin marks in human cells. *PLoS Genet.* 10:e1004282.
- Quinet A, Tirman S, Jackson J, Šviković S, Lemaçon D, Carvajal-Maldonado D, González-Acosta D, Vessoni AT, Cybulla E, Wood M, Tavis S, Batista LFZ, Méndez J, Sale JE, Vindigni A. 2020. PRIMPOL-Mediated Adaptive Response Suppresses Replication Fork Reversal in BRCA-Deficient Cells. *Mol Cell.* 77:461-74.e9.
- Ray Chaudhuri A, Hashimoto Y, Herrador R, Neelsen KJ, Fachinetti D, Bermejo R, Cocito A, Costanzo V, Lopes M. 2012. Topoisomerase I poisoning results in PARP-mediated replication fork reversal. *Nat Struct Mol Biol.* 19:417-23.
- Schürmann L, Schumacher L, Roquette K, Brozovic A, Fritz G. 2021. Inhibition of the DSB repair protein RAD51 potentiates the cytotoxic efficacy of doxorubicin via promoting apoptosis-related death pathways. *Cancer Lett.* 520:361-73.
- Simandlova J, Zagelbaum J, Payne MJ, et al. 2013. FBH1 helicase disrupts RAD51 filaments in vitro and modulates homologous recombination in mammalian cells. *J Biol Chem* 288:34168-80.
- Zellweger R, Dalcher D, Mutreja K, Berti M, Schmid JA, Herrador R, Vindigni A, Lopes M. 2015. Rad51-mediated replication fork reversal is a global response to genotoxic treatments in human cells. *J Cell Biol* 208:563-79.

Dr. Juan Mendez
CNIO
Molecular Oncology Programme
Melchor Fernandez Almagro, 3
Madrid E-28029
Spain

8th Jan 2024

Re: EMBOJ-2023-114022R
RAD51 restricts DNA over-replication from re-activated origins

Dear Juan,

A (belated) Happy New Year! And thank you for submitting your revised manuscript, which has now been seen once more by the two original referees. Since both were fully satisfied with your revisions and responses to the initial comments, we shall be happy to accept it for EMBO Journal publication, following incorporation of the following few editorial points:

- Please upload all main Figures and all Expanded View figures as individual files with sufficient resolution/quality for production. Please also move all Expanded View figure legends into the main text file, after the main figure legends.
 - Please double-check to make sure to all relevant funding information in the manuscript is congruent with the info entered into our submission system (currently mismatched: PID2019-106707-RB vs. PI2019-106707-RB; and missing info in eJP: ERDF funds from the EU; EMBO post-doctoral Fellowship; CNIO Friends post-doctoral contract).
 - As we are switching from a free-text author contribution statement towards a more formal statement based on Contributor Role Taxonomy (CRediT) terms, please remove the present Author Contribution section and instead specify each author's contribution(s) directly in the Author Information page of our submission system during upload of the final manuscript. See <https://casrai.org/credit/> for more information.
 - Finally, during routine pre-acceptance checks, our data editors have raised the following queries regarding figures, data, and legends; I would appreciate if you briefly answered to them in the cover letter of your final submission, and made the requested text modifications with changes/additions highlighted via the "Track changes" option, to facilitate our final checking.
1. Please define the annotated p values *** in the legend of figure EV 2d; as appropriate.
 2. Please indicate the statistical test used for data analysis in the legend of figure EV 6b.
 3. Please note that in figure EV 1c, there is a mismatch between the annotated p values in the figure legend and the annotated p values in the figure file that should be corrected.
 4. Please note that the box plots need to be defined in terms of minima, maxima, centre, bounds of box and whiskers, in the legends of figures 1f; 2a; 3a, d; EV 4e.
 5. Please note that the box plots need to be defined in terms of minima, maxima, bounds of box and whiskers, and percentile in the legends of figures EV 4a-c.
 6. Although 'n' is provided, please describe the nature of entity for 'n' in the legends of figures 1d-f, h; 2c-e; 4f; EV 1a-d; EV 2b-d, g; EV 4d-e."
 7. Please note that in figures 4a, e; the scale bar unit should be corrected from μM to μm .

I am therefore returning the manuscript to you for a final round of minor revision, to allow you to make these adjustments and upload all modified files. Once we will have received them, we should be ready to swiftly proceed with formal acceptance and production of the manuscript.

With kind regards,

Hartmut

*** PLEASE NOTE: All revised manuscript are subject to initial checks for completeness and adherence to our formatting

guidelines. Revisions may be returned to the authors and delayed in their editorial re-evaluation if they fail to comply to the following requirements (see also our Guide to Authors for further information):

9) Digital image enhancement is acceptable practice, as long as it accurately represents the original data and conforms to community standards. If a figure has been subjected to significant electronic manipulation, this must be clearly noted in the figure legend and/or the 'Materials and Methods' section. The editors reserve the right to request original versions of figures and the original images that were used to assemble the figure. Finally, we generally encourage uploading of numerical as well as gel/blot image source data; for details see: embopress.org/page/journal/14602075/authorguide#sourcedata

At EMBO Press, we ask authors to provide source data for the main manuscript figures. Our source data coordinator will contact you to discuss which figure panels we would need source data for and will also provide you with helpful tips on how to upload and organize the files.

Further information is available in our Guide For Authors:

In the interest of ensuring the conceptual advance provided by the work, we recommend submitting a revision within 3 months (7th Apr 2024). Please discuss the revision progress ahead of this time with the editor if you require more time to complete the revisions. Use the link below to submit your revision:

Link Not Available

Referee #1:

In the revised manuscript, the authors reasonably addressed all concerns that I had raised and the manuscript is now improved. I recommend this manuscript for publication in the EMBO journal.

Referee #3:

The authors have done an excellent job of revising this manuscript. They now provide new experimental evidence to support their claims. In my opinion, they have correctly addressed all the issues raised by the three reviewers.

All editorial and formatting issues were resolved by the authors.

Dr. Juan Mendez
CNIO
Molecular Oncology Programme
Melchor Fernandez Almagro, 3
Madrid E-28029
Spain

12th Jan 2024

Re: EMBOJ-2023-114022R1
RAD51 restricts DNA over-replication from re-activated origins

Dear Juan,

Thank you for submitting your final revised manuscript for our consideration. I am pleased to inform you that we have now accepted it for publication in The EMBO Journal.

With kind regards,

Hartmut
